# Hotspots for social and ecological impacts from freshwater stress and storage loss

Xander Huggins [1,2], Tom Gleeson [1,3 ✉], Matti Kummu [4], Samuel C. Zipper [5], Yoshihide Wada [6], Tara J. Troy [1] & James S. Famiglietti[2,7]

Humans and ecosystems are deeply connected to, and through, the hydrological cycle. However, impacts of hydrological change on social and ecological systems are infrequently evaluated together at the global scale. Here, we focus on the potential for social and ecological impacts from freshwater stress and storage loss. We find basins with existing freshwater stress are drying (losing storage) disproportionately, exacerbating the challenges facing the water stressed versus non-stressed basins of the world. We map the global gradient in social-ecological vulnerability to freshwater stress and storage loss and identify hotspot basins for prioritization ($n = 168$). These most-vulnerable basins encompass over 1.5 billion people, 17% of global food crop production, 13% of global gross domestic product, and hundreds of significant wetlands. There are thus substantial social and ecological benefits to reducing vulnerability in hotspot basins, which can be achieved through hydro-diplomacy, social adaptive capacity building, and integrated water resources management practices.

[1] Department of Civil Engineering, University of Victoria, Victoria, Canada. [2] Global Institute for Water Security, University of Saskatchewan, Saskatoon, Canada. [3] School of Earth and Ocean Sciences, University of Victoria, Victoria, Canada. [4] Water and Development Research Group, Aalto University, Espoo, Finland. [5] Kansas Geological Survey, University of Kansas, Lawrence, KS, USA. [6] International Institute for Applied Systems Analysis, Laxenburg, Austria. [7] School of Environment and Sustainability, University of Saskatchewan, Saskatoon, Canada. ✉email: tgleeson@uvic.ca

Humans and ecosystems, alongside hydrological and biogeochemical cycles, are a deeply coupled and global social-ecological system[1–3]. Social-ecological systems are complex adaptive systems formed by interactions and feedbacks between biophysical and social processes[4,5], and freshwater is fundamental for flourishing and resilient ecosystems, societies, and the larger Earth System[3,6–8]. Global freshwater storage and flows are dynamic with oscillations and persistent trends driven by the combined influence of human activity, climate change, and natural variability on sub-seasonal to multi-decadal timescales[9,10]. Human dominance over the water cycle is increasingly recognized[11] and the continued development of global hydrological models enables higher fidelity representations of human and climate change impacts on water resources[12]. However, the science of understanding the reciprocal impacts of freshwater stress and storage trends on humans and ecosystems remains in its infancy at the global scale. We argue that studying both directions of this coupled social-ecological system (i.e., social-ecological activity impacts on freshwater and freshwater impacts on social-ecological activity) is crucial to confronting global freshwater challenges, yet the latter has received considerably less attention. In this paper, we consider the potential for freshwater stress and storage loss to impact humans and ecosystems. We do this by synthesizing a subset of the few but critical global ecohydrological and sociohydrological datasets with freshwater storage, freshwater withdrawal, and streamflow datasets (see Supplementary Table 1).

We seek to build on the existing literature on global freshwater scarcity and security topics, which broadly address social and ecological impacts of freshwater-related stresses and hazards. We refer here to freshwater scarcity studies as those which evaluate the ratios of water use to streamflow and streamflow per capita, typically at the basin scale, e.g.,[13–16] and to freshwater security studies as those which integrate multidimensional indicators of physical, chemical, socioeconomic, and institutional factors and aggregate using grid-based, basin, or administrative discretization schemes, e.g.,[17–19]. While water scarcity assessments exclusively focus on freshwater stress, it is only one element of physical water security and thus only one component of water security assessments. Both approaches, however, have important limitations that constrain their ability to support specific conclusions and drive policy implementation regarding the impacts of freshwater-related hazards such as stress and storage loss.

For instance, freshwater scarcity assessments typically apply globally-consistent classification schemes which do not represent important spatial variations in social and ecological sensitivities and responses. A few holistic derivatives of freshwater scarcity assessments, such as the social water stress[20] and water poverty[21] indices, have only been evaluated at the national level. Alternatively, water security assessments consider water scarcity as just one of many input variables. These assessments typically aggregate multidimensional indicators of different aspects of water security, which can lead to similar water security outcomes with different input indicator combinations. As a result, water scarcity impacts become challenging to isolate from final water security assessment results. Furthermore, this aggregation approach does not consider interactions or relationships between elements of water security which are critical determinants of social-ecological system behavior[22].

In this paper, we combine the strengths of water scarcity and water security research, and address their limitations by integrating concepts from social-ecological systems research. We combine concepts from these fields to address the following core objectives of this study: (1) Asses the global co-occurrence of freshwater stress and freshwater storage trends at the basin scale. (2) Analyze the relationship between social adaptive capacity and ecological sensitivity indicators with freshwater stress and storage trends. (3) Derive the global gradient in social-ecological vulnerability to freshwater stress and storage trends by considering all indicators listed above, and identify hotspot basins as those with high vulnerability values with respect to the global distribution. (4) Evaluate current levels of integrated water resources management within hotspot basins. Basins, at various scales, are an increasingly used and particularly suitable geospatial unit of analysis for hydrologically-based social-ecological systems analysis[23]. In this study, all analyses are performed at a large basin scale ($n = 1204$, median area ~70,000 km²). Input data align to the year 2015 as best as possible, and data are summarized to the basin scale by computing the area-weighted basin average or within-basin sum, depending on the intensive or extensive nature of each dataset (see "Methods" section and Supplementary Information). See Box 1 for definitions of key terminology used in this paper.

## Results

**The global co-occurrence of freshwater stress and freshwater storage trends**. We mapped freshwater stress and trends in freshwater storage at the basin scale and analyzed the co-occurrence of these phenomena (Fig. 1).

Freshwater stress represents the state of demand-driven water scarcity[15] and is defined as the ratio of freshwater withdrawal to streamflow (Fig. 1a). Trends in freshwater storage, conversely,

---

**Box 1 | Key terminology as used in this paper. See Methods for further information**

**Freshwater stress:** The ratio of annual freshwater withdrawal (W) to annual streamflow (Q). We refer to basins with W/Q ≥ 10% as **stressed basins** and those with W/Q ≥ 40% as **highly stressed basins**.

**Freshwater storage trends:** Year-over-year trends in total freshwater storage based on satellite observations over the 2002–2016 time period. Total freshwater storage is a vertically aggregated measure of water storage that includes groundwater, soil water, surface water, canopy water, and ice and snow water equivalents where present. For simplicity, we refer to negative freshwater storage trends as **drying trends** or **storage loss** and positive trends as **wetting trends** or **storage gain**.

**Basin freshwater status:** An integrated indicator that combines normalized freshwater stress and normalized freshwater storage trends at the basin scale. High indicator scores are assigned to basins with co-occurring freshwater stress and drying trends. We refer to high freshwater status scores through status **severity**.

**Vulnerability:** The likelihood of society and ecosystems to experience harms due to exposure to freshwater stress and storage loss when considered together as a basin's freshwater status. This vulnerability definition is an application of Turner et al.'s generic definition[29]. Vulnerability is quantified using social adaptability, ecological sensitivity, and basin freshwater status indicators. Social adaptability and ecological sensitivity indicators are described in the text and Methods.

**Hotspot basin:** Highlighted basins that possess the greatest vulnerability scores. We identify hotspot basins to support their prioritization in global water resources and integrated management initiatives. Basins are considered hotspots if sorted into "high" and "very high" vulnerability classes following a categorical classification of the numerical vulnerability results.

---

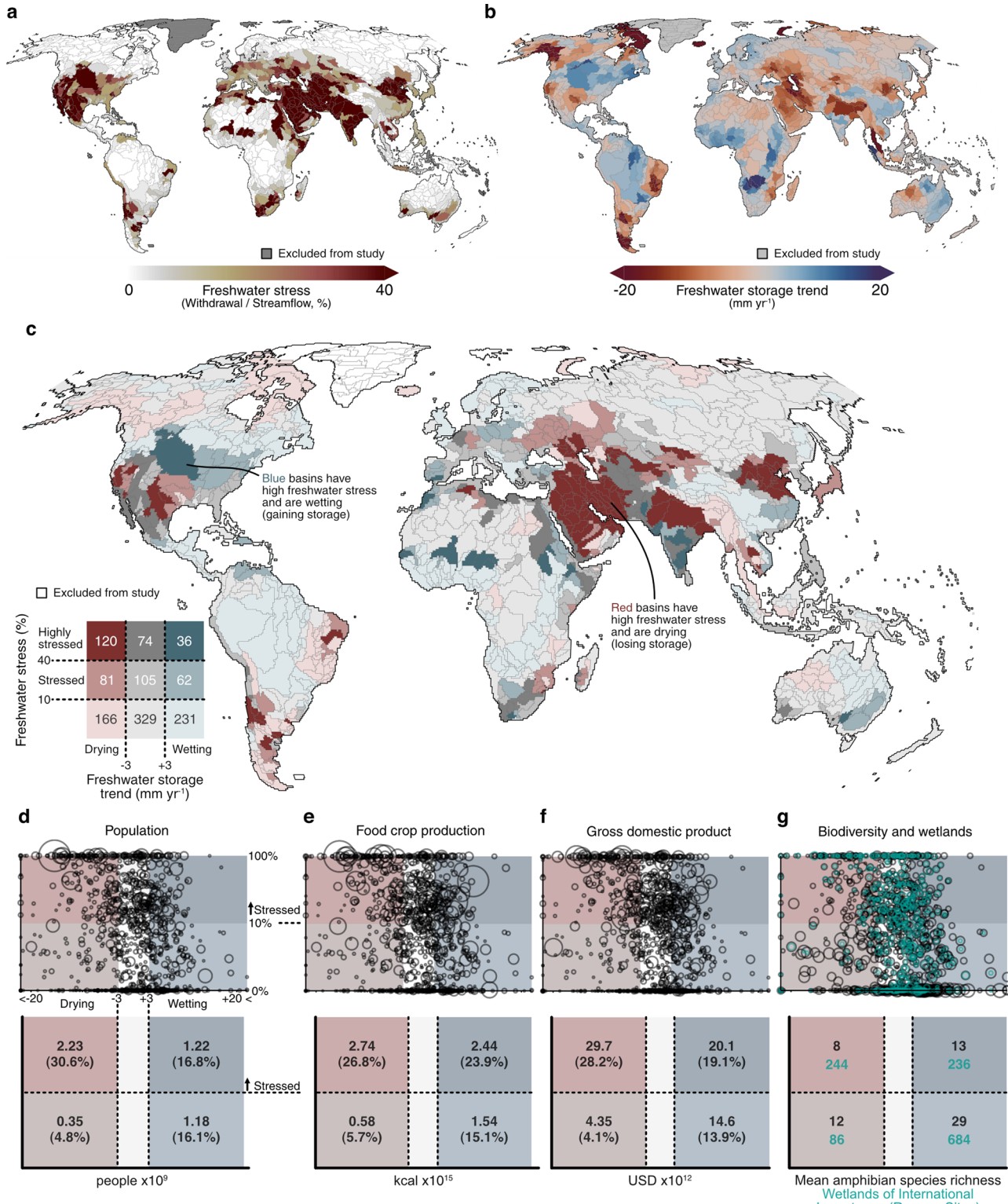

**Fig. 1 Global co-occurrence of freshwater stress and storage trends. a** Freshwater stress, derived from freshwater withdrawal and streamflow datasets (see "Methods" section). **b** Freshwater storage trend per basin. **c** Combinations of freshwater stress and storage trend per basin, which together derive basin freshwater status (shown in Fig. 2b). Values overlaying the legend indicate the number of basins satisfying each set of conditions. For categorical plotting purposes only, ±3 mm year⁻¹ is used as the threshold denoting a clear directional storage trend, based on the error level of the underlying observations[25]. **d–g** The exposure of social-ecological activity to freshwater stress and storage trends. Each plot represents storage trends as the *x*-axis coordinate, and log-transformed freshwater stress as the y-axis coordinate with the size of each circle based on the basin's value respective to each plotting dimension.

represent the evolution of total storage, defined as the vertical sum of groundwater, soil moisture, surface water, and snow water equivalent storages (Fig. 1b). Freshwater stress and storage are linked, as freshwater storage becomes a required source of water during periods when demands exceed supply. As climate change intensifies hydrological extremes globally, the strategic importance of the world's largest store of liquid freshwater, groundwater, will only continue to increase[24]. Though studies have focussed on global assessments of freshwater stress[13–15] and trends in freshwater storage[9], no study to date has mapped these two variables against one another. Doing so provides important context to differentiate basins of equal freshwater stress, as drying trends are likely to exacerbate challenges derived from freshwater stress, while wetting trends may yield offsetting effects. However, as freshwater stress calculations do not differentiate between withdrawals sourced from streamflow or storage, the two variables are not necessarily independent.

We found that 201 (42%) of the 478 currently stressed basins (withdrawal/streamflow > 0.10) are simultaneously losing freshwater storage (Fig. 1c). These basins are located in south and southwestern USA, northeastern Brazil, central Argentina, Algeria, and concentrate throughout the Middle East, the Caucasus, northern India, and northern China. Predominantly, these regions are agriculturally significant and heavily irrigated[9], with the exception of a few basins in South America whose trends are likely the product of natural variability[9]. Conversely, 98 (21%) of the currently stressed basins are gaining freshwater storage. The storage trends in these basins have largely been attributed to natural variability with the exception of central India, whose trends are partially attributed to groundwater recovery following groundwater policy change[9]. The remaining 179 stressed basins have freshwater storage trends that are smaller than can be definitively interpreted from the satellites monitoring these trends[25]. This skew towards negative storage trends (i.e., drying) in the world's water-stressed basins dissipates and even reverses in the non-stressed basins, where drying and wetting trends are found in 23% and 32% of the 726 non-stressed basins, respectively. While previous work has shown that the world's dry regions are becoming drier while the wet regions are becoming wetter[26], this work reveals that the stressed regions of the world are becoming drier while the non-stressed regions of the world have no clear overall trend in freshwater storage.

The encompassed human population, food crop production, gross domestic product (GDP), biodiversity, and wetlands enumerate the potential social-ecological impacts from the current state of global freshwater stress and storage trends. Around 2.2 billion people, 27% of global food crop production, and 28% of global GDP live, grow, and situate in freshwater stressed basins that are drying (Fig. 1d–f). These totals represent an upper limit as not all social and ecological activity within these basins will be affected by freshwater stress and storage loss, which will depend on local levels of adaptive capacity and ecological sensitivity[22] (our focus in the subsequent sections). Conversely, 1.2 billion people, 24% of global food crop production, and 19% of global GDP are found in stressed basins that are wetting. We find less taxonomic biodiversity in the freshwater stressed and drying basins, and greater biodiversity in unstressed and wetting basins. Roughly the same number of wetlands of international importance are found in stressed and drying basins as in stressed and wetting basins. While these totals represent the magnitude of potentially affected biodiversity and wetlands, taxonomic biodiversity is only one of many critical facets of biodiversity[27], and freshwater stress and storage trends are but two of many variables impacting global biodiversity[28]. Thus, we urge caution in interpreting the role of freshwater stress and storage in driving differences in these biodiversity distributions.

**The most vulnerable populations to freshwater stress and storage loss**. To better characterize social vulnerability, freshwater stress and storage loss must be placed in the context of social adaptability. We mapped and analyzed the co-occurrence of freshwater stress and storage trends with an existing global dataset of social adaptive capacity[23] summarized at the basin scale (Fig. 2). Social adaptive capacity (Fig. 2a), or adaptability, represents "the ability of the system to respond to disturbances"[29] and is derived based on input indicators of governance, economic strength, and human development. This consideration of social adaptability enables more representative estimates of social, agricultural, and economic activity that are vulnerable to the co-occurrence of freshwater stress and storage loss. To consider freshwater stress and storage loss together, we developed the basin freshwater status indicator (Box 1) where higher values indicate co-occurring freshwater stress and storage loss (Fig. 2b, see "Methods" section).

We found 73 basins to possess low levels of social adaptability and severe basin freshwater status (Fig. 2c). These basins concentrate in Northern, and Eastern Africa, the Arabian Peninsula, and Western, Central, and Southern Asia; although vulnerable basins are also found in northeast Brazil, Southern Africa, and northern China. These basins encompass approximately 1.2 billion people, 12% of global food crop production, and 6% of global GDP (Fig. 2d–f). Conversely, 119 and 49 basins are found to have similarly severe basin freshwater status yet have moderate or high levels of social adaptability, respectively. These basins are located in southwestern USA and Mexico, Chile and Argentina, the Arabian Peninsula, regions surrounding the Caspian Sea, western Australia, and the North China Plain.

These differences in social adaptability across basins with severe freshwater status (i.e., co-occurring freshwater stress and storage loss) raise important economic considerations. First, greater social adaptability likely coincides with greater technological and economic capacity to pursue development. This development may consume greater volumes of freshwater and drive basins towards greater levels of freshwater stress or storage loss, while simultaneously increasing institutional and technical capacity to cope with limited water resources. Furthermore, freshwater stress and storage loss are not certain to induce negative economic impacts on basins, and can lead to positive impacts if a region is able to leverage its comparative advantages (e.g., irrigation efficiency) among other stressed regions[30]. Second, the divergent economic situation facing basins with severe freshwater status is particularly evident on a per-capita basis. In severe freshwater status, low adaptability basins, there resides 17% of the global population yet only 6% of global GDP. Conversely, in severe freshwater status basins with moderate-and-greater social adaptability, there resides 14% of the global population and an outsized 18% of global GDP (Fig. 2d, f). It is thus paramount that global initiatives prioritize and link economic inequality with freshwater goals. One such example is Sustainable Development Goal (SDG) 6.4 ("reduce the number of people suffering from water scarcity"), which we argue should increasingly be linked to targets of SDG 10 ("reduce inequality within and among countries").

**Hotspot basins found on all continents**. We mapped the global gradient in social-ecological vulnerability to freshwater stress and storage loss at the basin scale and, from this, identified those with the greatest vulnerability as hotspot basins (Fig. 3). Hotspot mapping has been a successful endeavor within the field of conservation biogeography[31,32], and many global hydrology studies have identified regions of exceptional water scarcity and security challenges e.g.,[13–15,17–19]. Here, we seek to combine and apply these concepts in an integrated global social-ecological

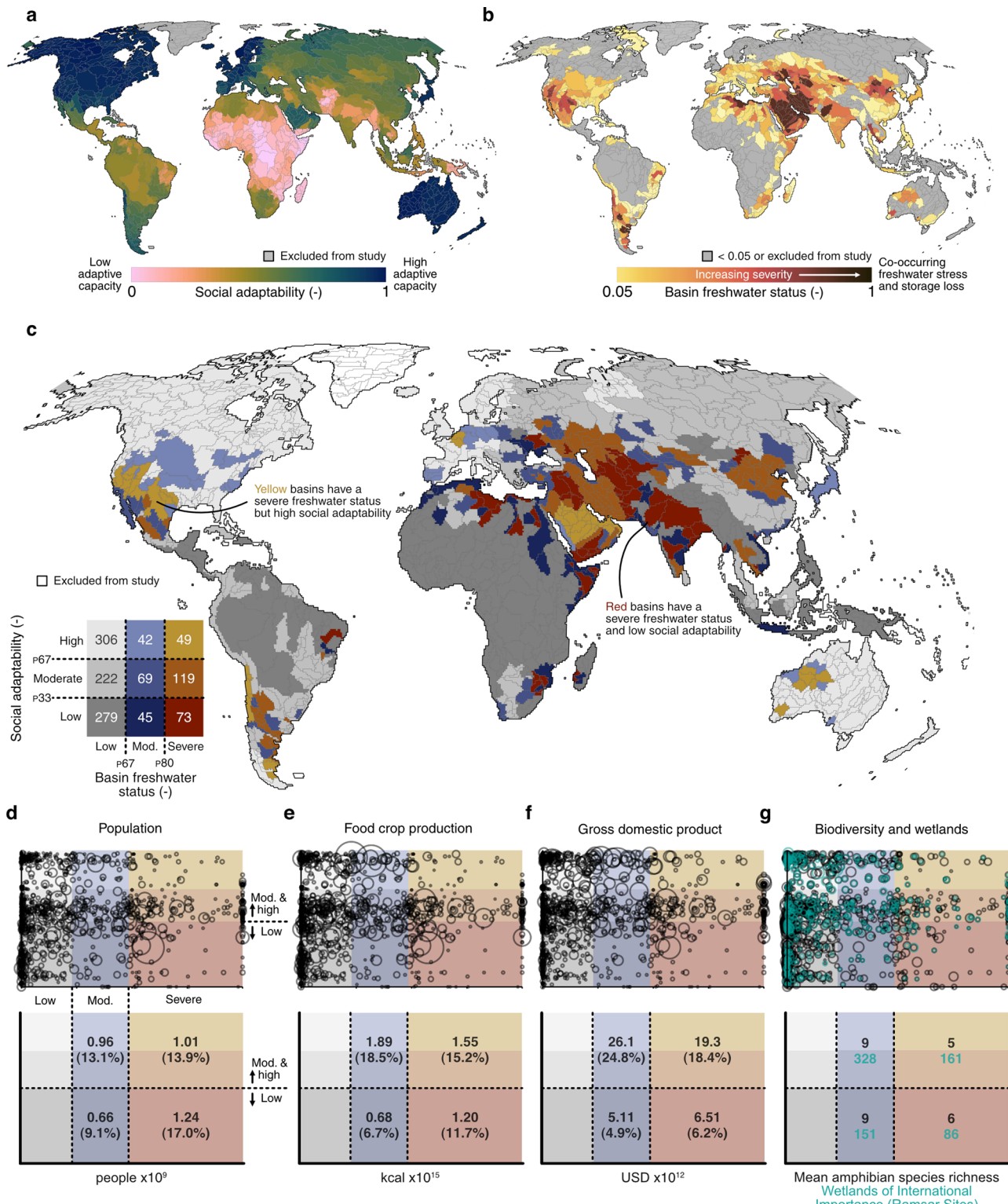

**Fig. 2 The relationship between basin freshwater status and social adaptive capacity. a** Social adaptive capacity, or adaptability, per basin. **b** Basin freshwater status, representing the combination of freshwater stress and storage trend per basin (see "Methods" section). **c** Combinations of basin freshwater status and social adaptability. Values overlaying the legend indicate the number of basins satisfying each set of conditions. **d**–**g** The exposure of social-ecological activity to basin freshwater status (*x*-axis coordinate) and social adaptive capacity (*y*-axis coordinate), with the size of each circle scaled based on the basin's value respective to each plotting dimension. These distributions are summarized below each plot. *P* notation represents the percentile distribution.

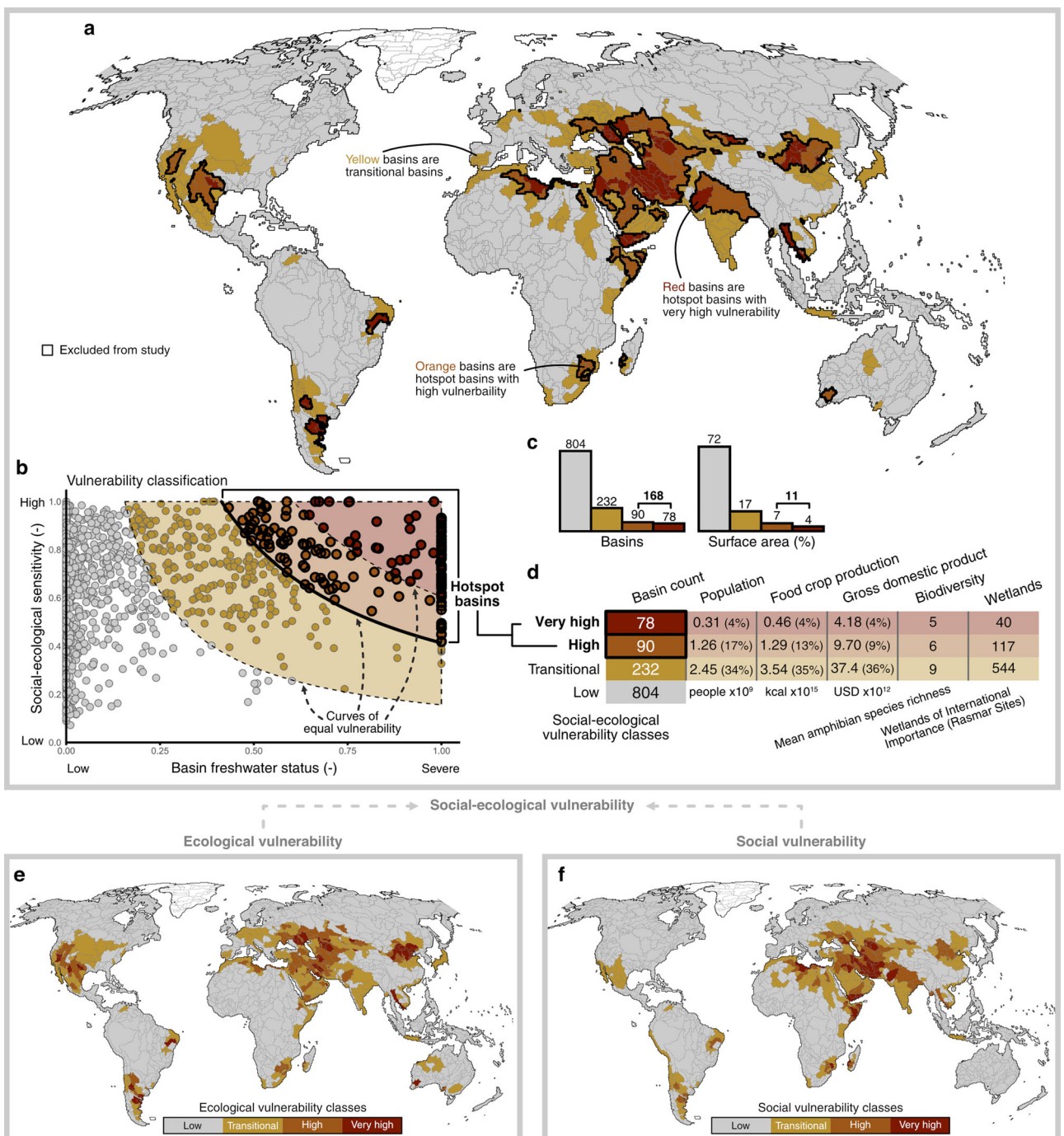

**Fig. 3 Hotspot basins for social and ecological impacts from freshwater stress and storage loss. a–d** Social-ecological vulnerability results. **a** Hotspot basins of social-ecological vulnerability to freshwater stress and storage loss. **b** Vulnerability classification, based on the product of basin freshwater status and social-ecological sensitivity to freshwater stress and storage loss (see "Methods" section). **c** Histograms of the global distribution of vulnerability classes by basin count and surface area. **d** Summarized social-ecological activity within transitional and hotspot basins. **e** Ecological vulnerability results, presented as vulnerability classes. **f** Social vulnerability results, presented as vulnerability classes. Vulnerability classes for **e** and **f** are derived using the same methods as shown for social-ecological vulnerability in **b**.

vulnerability context. As a useful reference, biodiversity hotspots aim to "maximize the number of species "saved" given available resources" by asking "where are places rich in species and under threat?"[33]. For comparison, the aim of our hotspot mapping is to 'minimize the social and ecological impacts of freshwater stress and storage loss given available resources' by asking "what basins with sensitive ecosystems and limited social adaptive capacity are exposed to freshwater stress and storage loss?"

We conceptualize vulnerability as the product of (i) ecological sensitivity, (ii) social adaptive capacity, and (iii) basin freshwater status. To represent ecological sensitivity, we derived an indicator using data products from two global ecohydrological studies that assess broad ecosystem sensitivity to freshwater storage and use (see "Methods" section). To represent social adaptability, we utilized the same adaptive capacity dataset as used in the previous section (Fig. 2a). To classify the derived global vulnerability

results into hotspot basins, we implemented a simple classification algorithm developed for heavy-tailed distributions[34], which appropriately describes the global vulnerability distribution.

The most vulnerable basins are constrained to regions confronting co-occurring freshwater stress and storage loss. When considering social and ecological vulnerability individually (Fig. 3e, f), we find spatial variation between ecological vulnerability (Fig. 3e) and social vulnerability (Fig. 3f). For instance, several basins in affluent nations with sensitive ecosystems reveal high ecological vulnerability but low social vulnerability (southwestern USA; western Australia). Conversely, several basins in Eastern Africa and northeastern India possess high social vulnerability but low to moderate ecological vulnerability. While these differences are notable and could impact regional strategies, it remains essential in most, if not all, regions that social and ecological vulnerabilities be confronted simultaneously[4]. For this purpose, we combined ecological sensitivity and adaptive capacity indicators into a combined social-ecological sensitivity indicator (see "Methods" section) to map combined social-ecological vulnerability (Fig. 3a).

We identify 168 basins, representing 14% of all basins and 11% of the global land area considered in our study, as vulnerability hotspots (Fig. 3a–c). These hotspot basins consist of basins receiving "high" and "very high" vulnerability scores through our classification procedure. Of the 168 basins, 78 (6% of all basins) are classified in the most-severe "very high" vulnerability class, while 90 (7% of all basins) are classified in the "high" vulnerability class. We also identified 232 basins (19% of all basins) as "transitional" basins, which are not classified alongside basins with null vulnerability yet also do not possess extreme values within the global vulnerability distribution. The 78 hotspot basins with "very high" vulnerability represent the multiple epicenters for potential social and ecosystem impacts from freshwater stress and storage loss. These basins are found in Argentina, northeastern Brazil, the American southwest, Mexico, Northern, Eastern, and Southern Africa, the Middle East and Arabian Peninsula, the Caucasus, West Asia, northern India and Pakistan, Southeastern Asia, and northern China.

A total of over 1.5 billion people, 17% of global food crop production, and 13% of global GDP are found within hotspot basins (Fig. 3d). Of these, ~300 million people, 4% of global food crop production, and 4% of global GDP situate within the 78 "very high" vulnerability basins. Consistent with the relationship between biodiversity and basin freshwater status, we find the most

vulnerable basins to be less taxonomically biodiverse than less vulnerable basins. While it is possible that these lower biodiversity levels may have eroded due to freshwater stress and storage loss, a proper investigation is outside the scope of this study and would require a wider array of pressures to be considered. The hotspot basins encompass 157 wetlands of international importance, which we highlight to prioritize their conservation in these vulnerable environments (Supplementary Table 2).

While the degree of social-ecological activity within hotspot basins is substantial, the global proportion of each dimension found in hotspot basins is roughly proportional to the fraction of basins within each vulnerability class. Thus, as the hotspot basins do not contribute disproportionately to global totals of social-ecological activity, we find it important to restate and clarify the motivating purpose of this hotspot mapping. The hotspot basins do not identify the greatest contributors to global social-ecological activity that face severe freshwater challenges. Rather, the hotspot basins are those with sensitive ecosystems and adaptability-limited societies exposed to the co-occurrence of freshwater stress and storage loss, and thus are the basins most likely to suffer social and ecological harms due to these freshwater conditions.

The identification of hotspot basins shows high levels of consistency across two uncertainty analyses and a sensitivity analysis focused on the impacts of subjective methodological decisions (Supplementary Section 4). We consider individually the impacts of (i) uniform over-estimation and under-estimation of each data input (spatially uniform uncertainty) and (ii) heterogeneous uncertainty in each data input (spatially variable uncertainty) on our hotspot basin results. Performing 10,000 realizations for each uncertainty analysis, we find that 98% of the identified transitional and hotspot basins are identified as at least transitional basins in over 50% the realizations considering spatially uniform uncertainty, and 96% when considering spatially variable uncertainty (Supplementary Figs. 8 and 9). The subjectivity-focused sensitivity analysis considered 24 alternative methodological configurations, and revealed that our identified transitional and hotspot basins are consistently identified across the majority of configurations (Supplementary Fig. 10).

**Implementation of integrated water resources management is inconsistent across hotspot basins.** We compared national implementation levels of integrated water resources management (IWRM) with our global vulnerability results (Fig. 4). For IWRM

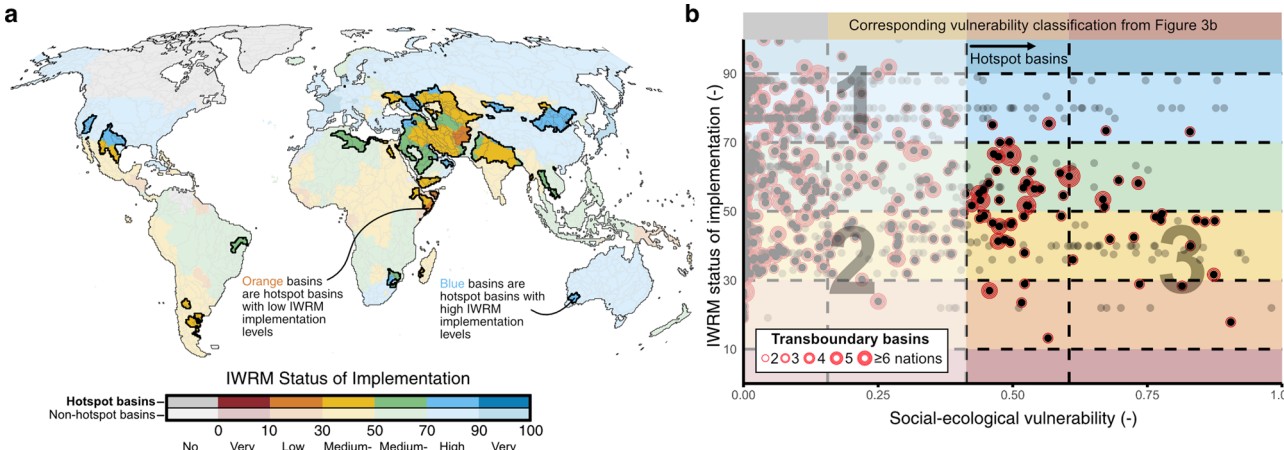

**Fig. 4 Integrated water resources management in hotspot basins. a** Map of IWRM implementation overlaid by hotspot basin results. **b** Scatterplot of individual basin values of social-ecological vulnerability (*x*-axis) and IWRM implementation (*y*-axis). Transboundary basins are represented by concentric red circles, with the number of circles representing the number of nations present within each basin. See text for interpretation of labels 1, 2, and 3.

implementation data, we rely on the IWRM Data Portal[35] which tracks progress on SDG 6.5.1 ("IWRM implementation at the national scale").

IWRM is defined as "a process which promotes the co-ordinated development and management of water, land and related resources, in order to maximize the resultant economic and social welfare in an equitable manner without compromising the sustainability of vital ecosystems"[36], while the SDG framework notes that IWRM implementation "supports all Goals across the 2030 Agenda"[37]. Thus, as the IWRM paradigm seeks to guide management of water resources to minimize trade-offs between human well-being, ecological health, and water resources sustainability, assessing implementation levels of IWRM against our vulnerability results provides insight regarding the performance of IWRM globally while simultaneously emphasizing the broad sustainability implications within hotspot basins.

Globally, we find no direct relationship between vulnerability and IWRM implementation at the basin scale. There is thus a wide range of IWRM implementation across all levels of social-ecological vulnerability to freshwater stress and storage loss, and there is no indication that IWRM implementation levels are greatest where they are most needed. This finding likely derives from variations in proactive versus reactive governance and management approaches to freshwater challenges across the globe. As our analysis is conducted at a snapshot in time (input data align to ~2015), we can only generate hypotheses about the performance of IWRM globally. For example, basins with high levels of IWRM implementation and low vulnerability (label 1 in Fig. 4b) have either proactively implemented IWRM, have effectively reduced their vulnerability through IWRM implementation, or simply benefit from a favorable intersection of regional climate and economy.

Alternatively, basins with low levels of IWRM and low vulnerability can be categorized as non-proactive in their IWRM implementation (label 2 in Fig. 4b). We place particular emphasis here on basins with low levels of IWRM where vulnerability is high (label 3 in Fig. 4b), which we argue should be the priority basins and regions of SDG 6.5-focused initiatives. Identified nations with low levels of IWRM implementation and very high vulnerability include Afghanistan, Algeria, Argentina, Egypt, India, Iraq, Kazakhstan, Mexico, Somalia, Ukraine, Uzbekistan, and Yemen. As one-third (36%) of all hotspot basins are transboundary (Fig. 4b), improving basin-level IWRM implementation will require multilateralism and hydro-diplomacy and cannot be left to individual nations acting alone. Furthermore, we observe a lower level of IWRM implementation across hotspot basins that are transboundary versus non-transboundary hotspot basins (mean basin IWRM Data Portal score = 50 vs. 56), suggesting greater multilateralism and cooperation are needed in transboundary basins.

## Discussion

**Implications for hotspot basins.** There are many possible social and ecological implications for hotspot basins, however these depend on basin-specific relationships among freshwater stress, storage trends, ecological sensitivity, and adaptive capacity. Ecologically, hotspot basins are more likely to suffer from transgressed environmental flows[38] with ensuing impacts on freshwater and riparian ecosystems[39]. Freshwater storage loss is linked to increased drought frequency[40], and falling water tables simultaneously decrease ecosystem resilience to drought by limiting root water-uptake[41], harming groundwater-dependent ecosystems[42], and perturbing the land surface energy balance[43]. Where freshwater stress and depletion are driven by irrigation,

the ecological implications are not limited to within the basin, as irrigation can modify moisture recycling and precipitation patterns across precipitationsheds[44], with cascading impacts on potentially distant ecosystems[45]. Socially, hotspot basins are forced to confront shrinking water resources to satisfy domestic, industrial, and agricultural demands[46,47]. Where water tables are dropping and wells are at risk of running dry[48], groundwater access may intensify social inequalities as only the wealthy may be able to afford to drill deeper wells[49]. Such conditions may trigger conflict and have implications on international security[28]. Of the near-700 water conflicts documented since 2000 by The Water Conflict Chronology[50], two-thirds (68%) are found within either transitional or hotspot basins from this analysis. As this study considers only a subset of water related stresses, this serves to indicate the potentially leading role freshwater stress and storage loss may play in instigating, contextualizing, and sustaining social conflict.

**Opportunities for global ecohydrology and sociohydrology.** This analysis is a representation of the best-available data for addressing global social-ecological impacts from freshwater stress and storage loss. However, it simultaneously highlights important limitations of existing ecohydrology and sociohydrology research at the global scale. For instance, we were required to develop an indicator to represent ecological sensitivity to freshwater from two existing ecohydrological datasets. However, these two datasets do not represent an exhaustive set of ecosystem processes and functions which may be impacted by changes in freshwater. We also do not address sub-grid variability considerations in either ecological sensitivity or social adaptability, which would independently benefit from specific study. Further, owing to a lack of alternatives, we adopted a rather reductionist and general derivation of social adaptive capacity as a proxy for social ability to respond and adapt to freshwater stress and storage loss. Future work that targets societal responses and relationships to specific freshwater stresses (e.g., Di Baldassarre et al.[51] but applied at the global scale and for various hydrological hazards) would be highly relevant for subsequent studies of this kind. We also only considered direct impacts of freshwater stress and storage loss, and thus did not consider indirect and non-local impacts such as the water and food security impacts of virtual water trade[52,53], which does not yet have data available globally at a sub-national scale.

In closing, we seek to establish a thematic connection between this work and Abbott et al.[11], who found human activity to be largely absent in water cycle diagrams and suggested this inaccuracy contributes to a "misunderstanding of global hydrology by policymakers, researchers and the public". In a similar spirit, we observe that social-ecological system impacts of hydrological change are under-considered in global hydrological studies and we argue this underrepresentation contributes to a lack of awareness or misunderstanding of ecohydrological and sociohydrological connections at the global scale. Addressing hydrological phenomena simultaneously as products and drivers of change within the global social-ecological system can only elevate the consideration freshwater will receive in complex, multi-objective, multi-disciplinary decision making. Freshwater stress and storage trends are only two of several critical aspects of freshwater with broad social-ecological sustainability and resilience implications[3]. Pending data availability, similar analyses can be performed for seasonal and inter-annual variability in water storage and several quality considerations. Developing such a network of studies will support a more comprehensive understanding of the social-ecological resilience implications of global hydrological change.

## Methods

**Study approach**. The overall study approach is summarized and illustrated in Supplementary Section 1 and Supplementary Figs. 1 and 2. Below, we focus on the specific methods performed in our analysis. A flow chart of our methodology is shown in Supplementary Fig. 3.

**Data selection**. All underlying data to this study were retrieved from pre-existing, published, and open datasets. We used 12 geospatial datasets and one basin scheme. We aligned our input data to the year 2015 as best as possible. We summarize input dataset selection, justification, and relevant preprocessing in Supplementary Table 1 (Supplementary Section 2).

**Geospatial methods**. We performed all analyses using the basin scheme of HydroBASINS level 4[54] and at the spatial resolution of 0.5 degrees using the World Geodetic Reference System 1984 ellipsoid (WGS 84). All raw input datasets were harmonized to 0.5 degrees using methods outlined in Supplementary Table 1. All data were summarized to the basin scale by (i) calculating the area-weighted average value of intensive properties within each basin using an algorithm to calculate cell area on a geographic grid[55], or (ii) calculating the within-basin sum of extensive properties. We masked basins in Greenland, northern Canada, and several small islands from our analysis due to inconsistent data coverage. This masking reduced the HydroBASINS level 4 discretization scheme to 1204 basins from an original set of 1341 (90% retention by count, 97% by surface area).

**Hotspot basins**. For a more extensive summary of the theory and justification of our approach, see Supplementary Section 1. In brief, we base our analysis on the vulnerability definition of Turner et al.[29] although there is considerable consensus around the general principles of vulnerability[56]. Our approach is similar to that of Varis et al.[23], a recent global assessment of river basin resilience using social-ecological principles. However, our study is of a narrower, more specific scope directed at the potential for social and ecological impacts from freshwater stress and storage loss rather than evaluating broad basin resilience to a wide range of ecological vulnerabilities.

We analyze social-ecological system vulnerability as the product of (i) exposure to freshwater stress and storage loss, represented by the basin freshwater status indicator, and (ii) the combination of ecological sensitivity and social adaptive capacity, which we represent using derived indicators.

Basin freshwater status is derived to represent exposure to the spatial co-occurrence and severity of freshwater stress and freshwater storage trends. This approach enables trends in freshwater storage to differentiate basins of equal freshwater stress as storage trends can aggravate or offset existing stress levels depending on the direction of the storage trend and the existing stress level. The derivation of this indicator is summarized in the "Basin freshwater status" section.

The social-ecological sensitivity indicator represents the ecological ability to either adapt or absorb freshwater stress or storage perturbations and social ability to adapt proactively and reactively to generic stresses[23]. The derivation of this social-ecological sensitivity indicator is summarized in the section below entitled "Ecosystem sensitivity and social adaptive capacity". We note that other common quantitative vulnerability approaches incorporate additional considerations such as scaling the sensitivity term by the proximity of the system state to a critical threshold, and undertaking a probabilistic approach to exposure[57]. Such thresholds, however, are highly uncertain, spatially-variable, or unknown in most complex adaptive systems[58], including human-water systems at the global scale[59]. Thus, we did not perform this threshold proximity scaling of the sensitivity term. Furthermore, as we evaluated the current state of freshwater stress and the existing trend in freshwater storage, there was no probabilistic component to our analysis. Vulnerability was thus represented in our analysis through Eq. (1).

$$V_i = S_i B_i, \tag{1}$$

where $V$ represents vulnerability of the social-ecological system, $S$ represents the combined social-ecological sensitivity indicator, and $B$ represents the basin freshwater status, per basin $i$.

**Basin freshwater status**. We derived basin freshwater status to compress the bivariate relationship of freshwater stress and storage trends into a single indicator. The indicator is a composite of both freshwater stress and storage trend inputs which we individually normalized using the value of 0.4 times annual streamflow ($Q$) per basin. The composite indicator (Eq. 4) is the arithmetic mean of the normalized freshwater stress indicator (Eq. 2) and the normalized storage trend indicator (Eq. 3). Our freshwater stress calculation differs from other freshwater stress studies which consider water sharing rules between upstream and downstream basins[15] and treat arid regions separately[60]. As these existing freshwater stress dataset are not available at our operating scale of HydroBASINS level 4, we calculated the ratio of withdrawal to streamflow within each basin and interpret this ratio as an approximate, relative measure of freshwater demand to within-basin generated renewable freshwater. We normalized both freshwater stress and freshwater storage trends by 0.4Q as this threshold is used throughout the freshwater stress literature to denote high basin stress levels[13,15,61,62]. Following common approaches in other global indicator-based assessments, we bound both normalized indicators to a maximum magnitude of 1, i.e., we set an upper limit of 1 for the normalized freshwater stress results by setting all values >1 to 1, and set upper and lower limits of 1 and −1 for the normalized storage trends by setting all vales <−1 to −1 and >1 to 1. We also "flipped" the normalized storage trend indicator (i.e., multiplied the indicator by −1) so that drying trends drying trends correspond to positive indicator scores for consistency with the freshwater stress indicator for which greater (more positive) values correspond with greater levels of stress. Basin freshwater status was calculated as the arithmetic mean of these two indicators, with a minimum value set to 0 as negative values are possible where wetting trends offset existing freshwater stress. Where large earthquakes interfered with storage trend observations (i.e., the 2011 Tohoku earthquake and the 2004 Sumatra-Andaman earthquake)[9], basin freshwater status was set to the independent normalized freshwater stress indicator alone. Input data for these indicators are shown in Supplementary Fig. 4.

$$F_i = \min\left(\frac{W_i}{0.4Q_i}, 1\right), \tag{2}$$

$$T_i = \max\left(\min\left(\frac{\frac{dTWS_i}{dt}}{0.4Q_i}, 1\right), -1\right) \times -1, \tag{3}$$

$$B_i = \max\left(\frac{F_i + T_i}{2}, 0\right), \tag{4}$$

where $F$ is the freshwater stress indicator, $T$ is the normalized freshwater storage trend indicator, and $B$ is basin freshwater status. $W$ represents annual freshwater withdrawals (mm year$^{-1}$), $Q$ represents annual streamflow (mm year$^{-1}$), and $\frac{dTWS}{dt}$ represents year-over-year trends in freshwater storage (mm year$^{-1}$), per basin $i$.

**Ecosystem sensitivity and social adaptive capacity**. We derived an indicator to represent general ecological sensitivity to freshwater stress and storage loss as no existing dataset fit this use. We combined data products from de Graaf et al.[63] and Seddon et al.[64], which represent the most relevant global ecohydrological studies. de Graaf et al. used a global surface water-groundwater model to estimate the groundwater head decline at which environmental flow limits are transgressed for all basins in which there is currently groundwater pumping. Seddon et al. developed the Vegetation Sensitivity Index to quantify the sensitivity in vegetation productivity to anomalies in three climate variables: water, temperature, and cloudiness, where water anomalies were represented by the ratio of actual evapo-transpiration to potential transpiration. We incorporated only the water-specific component of the Vegetation Sensitivity Index. In brief, the de Graaf et al. dataset represents the sensitivity of environmental flows to changes in groundwater storage in basins where groundwater is currently being withdrawn, while the Seddon et al. dataset represents vegetation sensitivity to anomalies in soil moisture and shallow groundwater storage.

To combine these ecohydrological datasets of different dimensions which simultaneously contribute to our overall understanding of ecological sensitivity to changes in freshwater storage, we performed a purely statistical, percentile-based approach. At the gridded resolution of 0.5 degrees, we converted both input datasets into area-weighted percentile datasets, where a value of 1 represents the grid cells with the global maximum values (99th–100th percentile) per dataset and values of 0.01 represent the grid cells with the global minimum values (0th–1st percentile) per dataset and where all values in-between apply to an equal proportion of the land surface. We then (i) computed the average value of both percentile-transformed datasets within each basin, (ii) averaged the two independent averages from (i) to combine the two datasets, and (iii) then normalized all basins by the global maximum basin value. This approach ensured that the most-sensitive basin received an ecosystem sensitivity indicator score of 1 and the least-sensitive basins received scores near 0. It is readily acknowledged, however, that a process-based derivation of ecosystem sensitivity that integrates groundwater, streamflow, and soil moisture considerations would be a superior alternative however none exist to our knowledge. The input data and derivation of the ecosystem sensitivity indicator are shown in Supplementary Fig. 5.

To reconcile the ecological sensitivity indicator (where higher values correspond with more sensitive basins) with the concept of social adaptive capacity, we inverted the dataset of social adaptive capacity so that greater values corresponded with lower adaptability. We sourced the adaptive capacity dataset from Varis et al.[23], as outlined in the main text. Social adaptive capacity is shown in Supplementary Fig. 6.

We combined ecosystem sensitivity and adaptive capacity indicators using the fuzzy sum operation (Eq. 4). The fuzzy sum is an increasing linear combination operator that ensures the sum is no less than the largest input value, yet the contribution of subsequent inputs decrease as inputs overlap. If all inputs are normalized [0, 1], the fuzzy sum converges on the upper limit (i.e., 1). The fuzzy sum is increasingly used in geospatial applications, such as in landscape integrity mapping[65] and we used the fuzzy sum as it eliminates the need to weight the ecosystem sensitivity and adaptive capacity inputs, which would introduce an additional degree of subjectivity. The fuzzy sum operation to derive the social-ecological sensitivity indicator is shown in Eq. (5). The resulting social-ecological

sensitivity indicator and its input datasets are shown in Supplementary Fig. 7.

$$S_i = 1 - (1 - E_i)(1 - (1 - A_i)), \tag{5}$$

where $S$ represents the social-ecological sensitivity indicator, $E$ represents ecological sensitivity, and $A$ represents social adaptive capacity, all per basin $i$. Note that the $(1 - A)$ term represents the inversion of the adaptive capacity dataset.

**Hotspot identification**. As our vulnerability analysis yields a relative gradient in global social-ecological vulnerability, classifying individual basins into vulnerability categories presents a particular challenge as no process-based thresholds are identified in the literature to differentiate the results. For example, while other prominent hotspot mapping initiatives such as the biodiversity hotspots are based on strict criteria (e.g., the biodiversity hotspots must contain ≥0.5% or ≥1500 of the world's plant species and must have lost ≥70% or more of its primary vegetation[32]), our social-ecological vulnerability hotspots are deeply interdisciplinary and multivariate, and thus make such explicit criteria challenging to identify. As vulnerability is not directly observable, the pragmatic approach is often to measure and classify relative vulnerability[57].

Given the heavy-tailed distribution of vulnerability results, we selected a relative classification algorithm, the Head/Tail Breaks method[34], to categorize the basins into vulnerability classes and subsequently hotspot basins. The Head/Tail Breaks classification scheme was developed to better represent the hierarchical structure of heavy-tailed distributions compared to other common methods such as Jenks natural breaks optimization. The classification scheme partitions the data into "head" and "tail" classes based on the arithmetic mean of the distribution and recursively re-partitions the "head" class based on the arithmetic mean of the "head" values until a skewness threshold is reached. For consistency between dimensions, we applied three iterations of the algorithm to all vulnerability distributions (i.e., social, ecological, and social-ecological) to partition the data consistently into four classes rather than use a skew-based stopping criterion. We classified the lowest-level class as non-hotspots (low vulnerability), the 2nd-level class as "transitional" basins (moderate vulnerability), and the 3rd-level and 4th-level classes as hotspots (high and very high vulnerability). As vulnerability is derived as the product of basin freshwater status and social ecological sensitivity, the class breaks can be represented by reciprocal function curves as shown in Fig. 3b (labeled as "Curves of equal vulnerability"). While implementing theory-based thresholds of vulnerability would be preferred, it is not realistic given current data availability and process knowledge of ecohydrological and sociohydrological systems at the global scale. We justify our approach as a data-driven, natural classification that characterizes the global gradient in social-ecological vulnerability to freshwater stress and storage loss.

**Uncertainty and sensitivity analyses**. We performed two uncertainty analyses to explore the impact on our hotspot mapping from potential input data uncertainty, and we performed one sensitivity analysis to identify the impact of subjective decision making in our methodology. These analyses are presented and described in Supplementary Section 4 and Supplementary Figs. 8–10.

## Data availability
The basin vulnerability and hotspot basin classification results from this study have been deposited in the University of Victoria's Scholars Portal Dataverse (https://doi.org/10.5683/SP3/SLR3GF). The raw data used in this study are accessible through several data and code repositories. We provide data sources, persistent web links, and descriptions of all data used in this study in Supplementary Table 1.

## Code availability
All analyses were conducted using the R project for statistical computing[66]. R packages necessary for analysis and visualization include: raster[67], sf[68], gdalUtils[69], spatstat.geom[70], ggplot2[71], tmap[72], and scico[73–75]. Composite figures were assembled in Affinity Designer (https://affinity.serif.com/en-us/designer/). The code generated to conduct this study is available in the GitHub repository https://github.com/XanderHuggins/HotspotBasins and is archived on Zenodo (https://doi.org/10.5281/zenodo.5728475)[76].

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

## Acknowledgements

We would like to thank colleagues at the Groundwater Science and Sustainability Research Group at the University of Victoria, and the Global Institute for Water Security at the University of Saskatchewan for reading and discussing earlier versions of this manuscript. X.H. was supported by the Natural Sciences and Engineering Research Council (NSERC) of Canada through an Alexander Graham Bell Canada Graduate Scholarship.

## Author contributions

X.H. conceived the study with advice from T.G., J.S.F., M.K., S.C.Z., Y.W. and T.J.T. X.H. assembled input data, performed the analysis, and wrote the manuscript with greatest input from T.G. and J.S.F. All authors, X.H., T.G., J.S.F., M.K., S.C.Z., Y.W. and T.J.T., discussed results and edited the manuscript at all stages.

## Competing interests

The authors declare no competing interests.
