## [Peer Review File · Nature Communications]

Hotspots for social and ecological impacts from freshwater stress and storage lossReviewer #1:

Remarks to the Author:

My full review is attached as a pdf. Thank you!

Reviewer Report for *Hotspots of social and ecological impacts from freshwater stress and storage loss*

Key results:

-The authors show the current state of global freshwater stress delineated by basin boundaries, rather than by national boundaries. This gives the reader a hydrological, process-based understanding of the problem and also a possible spatial (e.g., trans-national) approach to its solution.

-The maps produced in the study are multi-dimensional, combining indices of water storage, social vulnerability, and ecological vulnerability. As such, the authors present a matrix of a large gradient of combinations of vulnerability. From this, the authors identify the basins with the greatest vulnerability in all dimensions.

-The authors argue that these most vulnerable basins (across hydrological, social, and ecological dimensions simultaneously) are “hotspots” and are also deserving of global prioritization. The combined metrics of these basins are not trivial; these basins encompass over 1.5 billion people, 16 % of global crop calorie production, 14 % of global economic activity, and hundreds of significant wetlands.

-The results are robust, as the same hotspot basins are identified consistently across the majority of different methodological assumptions in the authors’ preliminary uncertainty and sensitivity analyses.

Significance of the work:

This work is novel in that it combines indices of large-scale hydrological systems with large-scale social governing systems, presenting a sophisticated justification for prioritizing potentially-vulnerable areas. As such, it could be a very valuable study for international hydro-diplomacy and also regional/international water resource management in general.

Major issues:

-There is a lack of clarity in how the work supports its conclusions. The manuscript reads as though nothing is described in a concrete way (e.g., authors refer to indices of indices, and inversions of other indices; the given definition of help provided by IWRM seems abstract and does not describe directly/literally how IWRM counterbalances the limitations/vulnerabilities that this manuscript tries to address).

-There is a lot of detail (actually, it could be pared down quite a bit) provided in the METHODS so that every keyword and index used in the work is defined. However, these original keywords

are used interchangeably, and differently, with their synonyms, which have definitions that are sufficiently different to be problematic, and this adds confusion. Often, the definitions and arguments appear to be and actually are circular.

-The following keywords (and their synonyms) need to be defined better (my thoughts are given below):

--**freshwater availability** (Line 16); “the combination of freshwater stress and trends in freshwater storage” (Line 56); “the global co-occurrence of freshwater stress and freshwater storage trends” (Line 62); “higher indicator levels correspond to decreasing freshwater availability (Line 135; But doesn’t “availability” include the “stress” component, the latter of which is calculated for the present time and is time-insensitive, so it cannot possibly decrease? Why not say “decreasing freshwater storage”? Also, if you “decrease freshwater availability,” are you decreasing the stress, or the storage?)”

There is a difference between freshwater availability and the engineered capacity to store and distribute water. Water “storage” and “availability” are used interchangeably, but storage is not necessarily available for use. Please clarify.

This manuscript does not distinguish between mechanisms causing decrease of freshwater availability (i.e., increase in freshwater stress). Is it a decrease of water input (e.g., drought in Line 314), or an increase of water output (e.g., anthropogenic water withdrawal mentioned in Line 149)? “Drying” in this manuscript is defined as the “loss of freshwater storage” (Line 86), but “drying” usually connotes the former (drought), not the latter (anthropogenic withdrawal), so the use of “drying” (e.g., Line 101) is ambiguous.

--**limited freshwater availability** (Line 27; 82)

--**freshwater scarcity** (Line 31); “ratios of water use to streamflow and streamflow per capita, typically at the basin scale (Line 33)”; “water scarcity assessments exclusively focus on freshwater stress” (Line 37); “support specific conclusions and drive policy implementation regarding the impacts of limited freshwater availability” (Line 40)

--**freshwater security** (Line 31); “water security assessments consider water scarcity as just one of many input variables”

--**freshwater stresses** (Line 32); “Freshwater stress represents the state of demand-driven water scarcity and is defined as the ratio of freshwater withdrawal to streamflow” (Line 71); “currently stressed basins (withdrawal/streamflow > 0.10)” (Line 85); “Freshwater stress, derived from freshwater withdrawal and streamflow datasets” (Line 115); “We developed an indicator to represent these two freshwater stresses, where higher indicator levels correspond to decreasing freshwater availability” (Line 134)

Basins with current freshwater stress may be drying, but that does not logically make them a water have-not region (they may have a larger volume of freshwater available to them overall), so it is not clear that there is a gap that is widening between water have and have-not regions.

--**social water stress index** (Line 45)

--**water poverty index** (Line 45)

--**freshwater storage** (Line 57); “Trends in freshwater storage, conversely, represent the evolution of total storage within each basin, defined as the vertical sum of groundwater, soil moisture, surface water, and snow water equivalent storages. Freshwater stress and storage are linked, as freshwater storage becomes a required source of water during periods where demands exceed supply” (Line 72); “Freshwater storage loss is linked to increased drought frequency” (Line 314)

How much freshwater storage loss must occur (what volume over what time period) for it to be classified as a loss or stress? Is water loss past, present, or future, because “loss” is used in all three ways throughout the manuscript.

--**social sensitivity to freshwater availability** (Line 58); “To consider social adaptability, we utilized the inverted adaptive capacity dataset” (Line 192)

--**ecological sensitivity to freshwater availability** (Line 58); “we derived an indicator from data products from two global ecohydrological studies that assess broad ecosystem sensitivity to freshwater availability variations or perturbations” (Line 190)

--**hotspot basins** (Line 66); “the aim of our hotspot mapping is to ‘minimize the social and ecological impacts of freshwater stress and storage loss given available resources’ by asking ‘where are basins with sensitive ecosystems and limited social adaptive capacity exposed to the combined threats of freshwater stress and storage loss?’” (Line 184; should this be moved up to the top of the manuscript before any results are given?); “To classify the global gradient in social-ecological vulnerability into hotspot basins, we implemented a classification approach developed for heavy-tailed distributions which was applicable to the global social-ecological vulnerability distribution” (Line 193); “social-ecological vulnerability hotspots to limited freshwater availability” (Line 210); “hotspots (i.e. basins with ‘high’ and ‘very high’ vulnerability)” (Line 214); “hotspot basins identify the basins with sensitive ecosystems and adaptability-limited societies exposed to the combined threats of freshwater stress and storage loss” (Line 237); “The inverse function curves represents curves of equal vulnerability, and differentiate basins into discrete vulnerability classes used for hotspot identification” (Line 258)

Why use the term “hotspot basin”? Define “hotspot”. “Spot” and “basin” seem at odds because a basin is by definition the largest hydrological unit (a basin is larger than a watershed) so it is hardly a spot; see Supplementary Material Line 171 (“we apply a coarse basin discretization scheme”). Authors might be referencing McClain et al. 2003 (“Biogeochemical Hot Spots and Hot Moments...”), but in that paper, “hotspots” are defined as “areas (or patches) that show disproportionately high reaction rates relative to the surrounding area (or matrix).” A basin has no surrounding area, as it encompasses all of the area of the hydrological unit.

With so many indicators of vulnerability and limitations, it is unclear what the hotspot refers to. Are the hotspots the basins which are unions of being stressed, having decreased storage trend, having low social adaptability, and having high ecological vulnerability? Or just one or a smaller combination of those? There are different categories of hotspots (e.g., ecological vs. social hotspot; Line 201)? Figure 3 shows ecological hotspots and social hotspots separately (Line 260)? Consider saying ecological-only or social-only hotspots to differentiate from “hotspot”, which refers to a specific subset of general hotspots. Do hotspots include “very high” and “high” vulnerability? This should be clarified at the first mention of “hotspot”.

--**drying trends** (Line 81); “losing freshwater storage (drying)” (Line 86)

--**wetting trends** (Line 82); “gaining freshwater storage (wetting)” (Line 89)

--**social vulnerability** (Line 127); “product of inverted social adaptability and the combined freshwater stress indicator represents social vulnerability” (Line 257)

--**social adaptability** (Line 128); “the ability of the system to respond to disturbances” and is derived based on input indicators of governance, economic strength, and human development” (Line 130); “a proxy for social ability to respond and adapt to freshwater stresses” (Line 340)

--**threat** (Line 134); “combined threats of freshwater stress and storage loss” (Line 134; isn’t “storage loss” = “drying”?)

--**combined stresses** (Lines 127, 138, 143)? What are these? Is it “drying” in a “stressed basin”? It is confusing when “freshwater stress” is one of the “combined stresses”. I see “combined freshwater stress” is defined further down in Line 147 as “co-occurring freshwater stress and storage loss”. Consider moving this definition up to its first use (Line 127).

There needs to be a different word for co-occurring freshwater stress and storage loss (must not be “freshwater stress” because it is circular; Line 147); even more confusing, “freshwater stress indicator” is defined by the relationship between freshwater stress and storage trends.

--**economic inequality** (Line 158); is this inequality within or among nations? Move up definition from Line 162?

--**vulnerability** (Line 188); “the product of (i) ecological sensitivity, (ii) inverted social adaptive capacity, and (iii) the combined freshwater stress indicator” (Line 188)

--**ecological vulnerability** (Line 257); “product of ecological sensitivity and the combined freshwater stress indicator represents ecological vulnerability” (Line 256)

--**social-ecological system vulnerability** (Line 382); “We analyze social-ecological system vulnerability as the product of (i) exposure, (ii) ecological sensitivity, and (iii) social adaptive capacity all with respect to freshwater stress and freshwater storage trends. Sensitivity and adaptive capacity are often considered separately in vulnerability assessments however the two

concepts are highly interlinked, and we consider the two together as it supports our focus on evaluating the coupled social and ecological vulnerability of basins.” (Line 382); “We derive social-ecological vulnerability to freshwater stress and storage loss as the product of (i) exposure to freshwater stress and storage loss and (ii) the combination of inverted social adaptive capacity and ecological sensitivity to hydrological disturbance, both of which are represented by derived indicators”; “Social-ecological vulnerability is thus represented in our analysis through equation (1)...” (Line 415)

--**resilience** (Line 388); “resilience, which can be illustrated by the feedback effects of social and ecological impacts of freshwater stress and storage trends that may alter social and ecosystem functioning and/or well-being” (Line 388)

--**exposure indicator** (Line 399); “the spatial co-occurrence and severity of freshwater stress and trends in freshwater storage” (Line 399)

--**social-ecological sensitivity indicator** (Line 404); “the ecological ability to either adapt or absorb freshwater availability perturbations, and social ability to adapt proactively and reactively to generic stresses” (Line 404)

-Some terms are defined by precise thresholds (freshwater stress is defined by a ratio of withdrawal/streamflow > 0.1) and equations (social-ecological system vulnerability is defined by Equation 1; Line 415), while others are not. These thresholds and equations should be made clear earlier, possibly at first mention of the term. Does “drying” have a threshold (perhaps a given percentage of water volume over a given period of time)?

-Describe time windows of all “trends”; are they past, past-up-to-present, or anticipated future trends?

-What is the global coverage of basins in this study? How many basins (by count and area) were and *were not* represented in this study relative to the total basins in the world (provide ratios); Line 375?

-What are the actual impacts of limited freshwater availability? Why does analyzing indices of freshwater availability matter? Please elaborate more and in a concrete way.

-The authors discuss social-ecological systems throughout the manuscript, but they present figures delineated only by hydrological system boundaries. Given that this manuscript centers largely around maps (all figures are world maps) and the authors chose to use this uncommon delineation of space by hydrological basin boundaries rather than by national borders or ecosystem boundaries, the authors need to be clearer about how they converted national and ecological statistics into basin statistics. If figures show basin boundaries that cross national boundaries, then how is, for example, social adaptability calculated within a hydrological basin boundary and also across a national boundary? Some of the answer is mentioned in the METHODS but it should be made clearer in the main text.

Line items:

Line 21: Change “enable” to “enables”; delete “greater”

Line 68: This section is titled “The global co-occurrence of freshwater stress and water availability trends”, but (fresh) water availability was just earlier defined as “the global co-occurrence of freshwater stress and freshwater storage trends” (Line 62), so “freshwater stress” is redundant (i.e., it is named explicitly and then again implicitly in “water availability”). Why not title this section “The global freshwater availability trends” or “The global co-occurrence of freshwater stress and freshwater storage trends”?

Line 76: Change “where” to “when”

Line 96: “by-and-large” is too informal

Line 115: The figure caption is titled “Global co-occurrence of freshwater stress and trends in storage”, so again there is a problem similar to in Line 68. Why not simplify to “Global freshwater availability trends”?

Line 246: 98 or 97 %?

Line 259: “represents” should be “represent”

Line 285: Move “(input data align to ~2015)” up earlier to where input data are first described

Line 303: There are basins in the figure that look gray and shaded (e.g., most of North America; India). This is confusing because in Fig 4a, the caption says “Gray basins are hotspot basins with no data on IWRM implementation levels” but in the scale bar it says “Non-hotspots are shaded to emphasize hotspots”.

Line 315: Is this a tautology? Generally, resilience is defined as success in the face of hardship/limitation, so here resilience should be agnostic to the drying conditions. Drying conditions should not decrease ecosystem resilience to drought. See definition in Supplementary Material Line 30 and 37. There, resilience is defined outside of the perturbation (e.g., drying conditions).

Line 324: Conflict is pervasive and inevitable, so its triggering is not that meaningful. Is there persistent or violent conflict linked to freshwater stress? If so, that would be worth mentioning.

Line 346: Wording? How about “whose data currently exist delineated by national jurisdiction rather than basin boundaries”?

Line 358: “analysis” should be “analyses”

Supplementary Material

Line items:

Line 42: There are three grey boxes and two shades of grey. It is “grey” here but “gray” in Fig. 4.

Line 94: Add “to” after “(increases)”

Line 145: “feedback” is two words

Line 147: How does climate change result in increasing rates of freshwater demand? Is it more plausible that climate change results in decreasing freshwater supply while population increase results in increasing freshwater demand? Also, climate change (as a significant driver of loss of freshwater storage) is not discussed in the main body of the manuscript, so either discuss climate change more in the main body, or less in the Supplementary Material.

Table 1: Crop calorie production. The temporal range of this data set is 1997–2003 but the target year of analysis is 2015. This may result in a serious underreporting of crop production. The world population in 2003 was 6.4B, whereas the world population in 2015 was 7.4B (an increase of 16%). Can you scale your crop calorie production to reflect this increase in calorie demand?

Line 220: “Bain” should be “Basin”

Reviewer #2:

Remarks to the Author:

I was VERY excited when I started reading this paper because of the multidimensional approach to water scarcity and social vulnerability and also because it incorporated remotely sensed hydrology. I think the approach is clever and innovative. I was actually quite surprised (and a little disappointed) when the results of this clever, multidimensional approach highlighted essentially the same regions as previous studies (over the last couple of decades) have found using the much simpler metrics (like demand/supply). But of course, this makes sense because all this vulnerability is likely due to human water demand and overuse of water resources. However, the additional information about how water scarcity, storage trends and social vulnerability indicators overlap with population, agriculture, economic activity and biodiversity adds substantially to that story and is an important finding.

Where this paper becomes truly IMPORTANT is when the authors evaluate IWRM efforts across the globe and assess how IWRM efforts overlap with hotspot basins. This is BRILLIANT and an incredibly important finding. This paper should be both motivational and useful to global efforts to advance IWRM.

The paper is innovative, well written and important; it should be published. I only have one tiny remark: the statement in lines 179 through 181 is really not true. Vorosmarty and collaborators, and other groups, have published many papers that identify hotspots of water scarcity and water insecurity issues. I would delete, or at least modify, this statement.

We thank the reviewers for their thoughtful and useful comments on our manuscript. We have made several modifications to the manuscript, largely centered around clarifying our use of terminology. All text changes have been tracked in the revised manuscript.

In summary, the following changes have been made to the manuscript in response to reviewer comments:

1. We have clarified our use of terminology substantially and added a box defining key terminology as used in the manuscript (Box 1 in main text).
2. To better align our crop production data with the year 2015, we replaced the crop production dataset used in the original submission (from Cassidy et al. 2013 with data for the year 2000) with Kummu et al. (2021)'s food crop production dataset for the year 2010.
3. We pared down the Methods and Supplementary Information to improve readability.
4. We have made a number of edits to the text and figures to reflect the changes listed above in addition to line-specific comments made by the reviewers.

In addition to the edits made in response to the reviewer comments, we made three other changes to improve the manuscript and analysis, which we would like to note:

1. We modified the resampling method performed on the adaptive capacity dataset in the 'data preprocessing' stage to make our resampling methods consistent across all data inputs. This change (from bilinear resampling to a calculated mean of all contributing cells) has a minor effect on our hotspot basin results, changing the total of hotspot basins from 172 to 168, however the spatial patterns and conclusions of our work remain unchanged.
2. The IWRM Data Portal released new data on IWRM implementation levels for the year 2020 since our original submission. As several notable countries were not included in the original IWRM dataset (e.g. USA, India), we have updated the IWRM data with this latest release.
3. We identified a minor error in our code used to report the summary statistics for Figure 2g-d. We have now fixed this error and corrected all summary statistics within the figure and all references to them in the text.

Our responses to individual reviewer comments are in blue below.

Reviewer 1

Reviewer Report for Hotspots of social and ecological impacts from freshwater stress and storage loss

We thank the reviewer for their thorough, critical, and helpful feedback on our manuscript.

As the reviewer's "Key results" and "Significance of the work" sections are descriptive and non-critical, we do not provide point-by-point responses to these sections.

Key results:

- The authors show the current state of global freshwater stress delineated by basin boundaries, rather than by national boundaries. This gives the reader a hydrological, process-based understanding of the problem and also a possible spatial (e.g., trans-national) approach to its solution.
- The maps produced in the study are multi-dimensional, combining indices of water storage, social vulnerability, and ecological vulnerability. As such, the authors present a matrix of a large gradient of combinations of vulnerability. From this, the authors identify the basins with the greatest vulnerability in all dimensions.
- The authors argue that these most vulnerable basins (across hydrological, social, and ecological dimensions simultaneously) are “hotspots” and are also deserving of global prioritization. The combined metrics of these basins are not trivial; these basins encompass over 1.5 billion people, 16 % of global crop calorie production, 14 % of global economic activity, and hundreds of significant wetlands.
- The results are robust, as the same hotspot basins are identified consistently across the majority of different methodological assumptions in the authors’ preliminary uncertainty and sensitivity analyses.

Significance of the work:

This work is novel in that it combines indices of large-scale hydrological systems with large-scale social governing systems, presenting a sophisticated justification for prioritizing potentially-vulnerable areas. As such, it could be a very valuable study for international hydro-diplomacy and also regional/international water resource management in general.

Major issues:

- There is a lack of clarity in how the work supports its conclusions. The manuscript reads as though nothing is described in a concrete way (e.g., authors refer to indices of indices, and inversions of other indices; the given definition of help provided by IWRM seems abstract and does not describe directly/literally how IWRM counterbalances the limitations/vulnerabilities that this manuscript tries to address).

We thank the reviewer for raising concerns around clarity of the manuscript, and have revised the manuscript thoroughly to address this. Many of our edits in response to subsequent comments below should support improved clarity, however we would like to take this opportunity to discuss the issue raised by the reviewer on how our work supports its core contribution.

The central contribution of this work is the identification of a set of basins that are the most vulnerable, globally, to experience social and ecological impacts from freshwater stress and storage loss. We document the derivation of these hotspot basins throughout the manuscript, from mapping the co-occurrence of freshwater stress and storage loss (Figure 1), to mapping the relationship between freshwater stress and storage loss with social adaptive capacity (Figure 2), to deriving social-ecological vulnerability to freshwater stress and storage loss and classifying these results to identify hotspot basins (Figure 3).

However, what we believe the reviewer may be alluding to, is the reliance on indicators in our methodology and the implications of this on the specificity of our discussion section (including on integrated water resources management, IWRM). Indeed, this study is indicator-based (as it may need to be, given the highly interdisciplinary nature of the driving question: ‘can we identify the basins of the world that are most vulnerable to social and ecological impacts from freshwater stress and storage loss?’). As we state in the main text and supplementary information, there is a lack of process-based knowledge on human-water and eco-hydrological interactions at the global scale, especially at the systems-level of analysis. Thus, our work is necessarily conceptual rather than process-based, i.e. our approach identifies general conditions that increase social and ecological vulnerability to freshwater stress and storage loss. As human and ecological responses to freshwater stress and storage loss will vary around the world, our discussions are thus necessarily general. For instance, as IWRM is a multifaceted paradigm of water management (for example, it is evaluated by the IWRM Data Portal using 33 sub-indicators), it is challenging to discuss the many mechanisms through which facets of IWRM can reduce the vulnerability our analysis highlights while simultaneously being sensitive to the reality that interventions and solutions to address vulnerability need to be locally attuned and context dependent. On this basis, we believe a more rigorous consideration of IWRM interventions to address social-ecological vulnerability is better suited to a separate, dedicated manuscript than to be included in this work. However, to address this reviewer comment, we have added extra justification for our comparison of IWRM implementation with hotspot basin results:

Line 279--:

“Thus, as the IWRM paradigm seeks to guide management of water resources to minimize trade-offs between human well-being, ecological health, and water resources sustainability, assessing implementation levels of IWRM against our vulnerability results provides insight regarding the performance of IWRM globally while simultaneously emphasizing the broad sustainability implications within hotspot basins.”

- There is a lot of detail (actually, it could be pared down quite a bit) provided in the METHODS so that every keyword and index used in the work is defined. However, these original keywords are used interchangeably, and differently, with their synonyms, which have definitions that are sufficiently different to be problematic, and this adds confusion. Often, the definitions and arguments appear to be and actually are circular.

We thank the reviewer for this feedback, and we are in complete agreement. We revisited our use of terminology throughout the manuscript to remove instances of interchangeably- used terminology or terms with unclear definitions. For example, in our original submission we problematically used the term “freshwater availability” to refer to freshwater storage, and we used “limited freshwater availability” to refer to the combination of freshwater stress and freshwater storage loss -- with the effect of confusing the reader on the meaning of the term “availability”. In our revised manuscript: (1) we do not use the term “freshwater availability” at all, (2) we refer to “freshwater stress” and “freshwater storage loss” as so explicitly, and (3) we refer to the combination of freshwater stress and freshwater storage loss as “basin freshwater status”.

We have added a box to the main text (Box 1) that provides definitions for key terminology as used in the work, which we believe will be a useful addition to the text. We also have pared down the Methods and Supplementary Information Section 1, as the reviewer suggested.

As the reviewer listed all terminology used in the manuscript below, we provide specific responses to each term regarding how each term was modified, removed, or retained in the manuscript.

- The following keywords (and their synonyms) need to be defined better (my thoughts are given below):

--**freshwater availability** (Line 16); “the combination of freshwater stress and trends in freshwater storage” (Line 56); “the global co-occurrence of freshwater stress and freshwater storage trends” (Line 62); “higher indicator levels correspond to decreasing freshwater availability (Line 135; But doesn’t “availability” include the “stress” component, the latter of which is calculated for the present time and is time-insensitive, so it cannot possibly decrease? Why not say “decreasing freshwater storage”? Also, if you “decrease freshwater availability,” are you decreasing the stress, or the storage?)”

As noted above, we have removed all use of the term “freshwater availability” from the manuscript and thank the reviewer for highlighting this source of confusion. Indeed, “availability” was being used inconsistently. We refer to freshwater storage trends exclusively as ‘freshwater storage trends’ in the revised manuscript to avoid such confusions.

There is a difference between freshwater availability and the engineered capacity to store and distribute water. Water “storage” and “availability” are used interchangeably, but storage is not necessarily available for use. Please clarify.

Indeed, there is an important distinction between freshwater availability and storage despite these terms occasionally being conflated in the literature -- though we might describe this distinction as that between freshwater availability and freshwater accessibility (see Damkjaer and Taylor 2017). However, as we have revised the manuscript to remove all uses of the term “freshwater availability”, we believe this removes the potential for this misrepresentation.

Reference:

Damkjaer, S., Taylor, R. The measurement of water scarcity: Defining a meaningful indicator. *Ambio* **46**, 513–531 (2017). <https://doi.org/10.1007/s13280-017-0912-z>

This manuscript does not distinguish between mechanisms causing decrease of freshwater availability (i.e., increase in freshwater stress). Is it a decrease of water input (e.g., drought in Line 314), or an increase of water output (e.g., anthropogenic water withdrawal mentioned in Line 149)? “Drying” in this manuscript is defined as the “loss of freshwater storage” (Line 86), but “drying” usually connotes the former (drought), not the latter (anthropogenic withdrawal), so the use of “drying” (e.g., Line 101) is ambiguous.

Indeed, the original manuscript did not attribute drivers to the trends in freshwater storage nor to the state of freshwater stress. This was done primarily on the basis that such analysis would be sufficient for a separate, dedicated analysis. Furthermore, drivers of freshwater storage trends have already been attributed by Rodell et al. (2018) to either natural variability, climate change, and/or direct human activity. However, in response to this comment, we have now added two sentences to the main text referring to the attribution of these drivers to freshwater storage trends, as identified by Rodell et al. - please see quotations below.

Regarding the use of the term “drying” to refer to loss of freshwater storage, we appreciate the argument of the reviewer that drying may be more commonly associated with drought than anthropogenic withdrawal. However, we also note that use of the term drying is also widely used in the context of human-driven storage loss. Thus our preference is to keep the use of the term drying as is, in reference to freshwater storage loss regardless of trend driver, which we believe improves the readability of the manuscript. However, accounting for the confusion raised by the reviewer through this comment, we have included “drying” and “wetting” in the key terminology box (Box 1).

Line 92--:

“These regions largely coincide with regions whose freshwater storage trends have been attributed to human activity, with the exception of those in South America whose trends are likely the product of natural variability⁹.”

Line 96--:

“The storage trends in these basins have largely been attributed to natural variability with the exception of central India whose trends are partially attributed to groundwater recovery following groundwater policy change⁹.”

Box 1 (within ‘Freshwater storage trends’ definition); beginning on Line 69:

*“For simplicity, we refer to negative freshwater storage trends as **drying trends** or **storage loss** and positive trends as **wetting trends** or **storage gain**.”*

Reference:

Rodell, M., Famiglietti, J.S., Wiese, D.N. et al. Emerging trends in global freshwater availability. *Nature* **557**, 651–659 (2018). <https://doi.org/10.1038/s41586-018-0123-1>

--limited freshwater availability (Line 27; 82)

As we have removed all instances of “freshwater availability” from the manuscript, “limited freshwater availability” is also no longer found. In the original submission, we had used the term “limited freshwater availability” to refer to the combination of freshwater stress and freshwater storage trends. Please see below and our response to “--combined stresses” for the new term we introduce that represents co-occurring freshwater stress and storage trends (“basin freshwater status”).

Box 1 (“Basin freshwater status” definition); beginning on Line 69:

“Basin freshwater status: An integrated indicator that combines normalized freshwater stress and normalized freshwater storage trends at the basin scale. High indicator scores are assigned to basins with co-occurring freshwater stress and drying trends. We refer to high freshwater status scores through status severity.”

--**freshwater scarcity** (Line 31); “ratios of water use to streamflow and streamflow per capita, typically at the basin scale (Line 33)”; “water scarcity assessments exclusively focus on freshwater stress” (Line 37); “support specific conclusions and drive policy implementation regarding the impacts of limited freshwater availability” (Line 40)

As our work builds on core concepts from the freshwater scarcity literature, the term “freshwater scarcity” is found in our manuscript as we provide a brief summary of the relevant core concepts in the introduction (see quotation below). We believe this is the appropriate location and extent to include this term in the manuscript, however as it is not used in the analysis sections, we do not include the term in our key terminology box (Box 1).

Line 32--:

“We seek to build on the existing literature on global freshwater scarcity and security topics which broadly address social and ecological impacts of freshwater-related stresses and hazards. We refer here to freshwater scarcity studies as those which evaluate the ratios of water use to streamflow and streamflow per capita, typically at the basin scale e.g. 13-16 .”

--**freshwater security** (Line 31); “water security assessments consider water scarcity as just one of many input variables”

Similarly to freshwater scarcity, our work also builds on core concepts and methods in freshwater security research. Thus, we include a brief overview of freshwater security in the introduction as necessary context of this work (see quotations below). We believe this is the appropriate location and extent to include this term in the manuscript, and likewise to freshwater scarcity, we do not include it in our key terminology box (Box 1).

Line 35--:

“... and to freshwater security studies as those which integrate multidimensional indicators of physical, chemical, socioeconomic, and institutional factors and aggregate using grid-based, basin, or administrative discretization schemes e.g. 17-19 .”

--**freshwater stresses** (Line 32); “Freshwater stress represents the state of demand-driven water scarcity and is defined as the ratio of freshwater withdrawal to streamflow” (Line 71); “currently stressed basins (withdrawal/streamflow > 0.10)” (Line 85); “Freshwater stress, derived from freshwater withdrawal and streamflow datasets” (Line 115); “We developed an indicator to represent these two freshwater stresses, where higher indicator levels correspond to decreasing freshwater availability” (Line 134)

See comment provided for “--combined stresses” below.

Basins with current freshwater stress may be drying, but that does not logically make them a water have-not region (they may have a larger volume of freshwater available to them overall), so it is not clear that there is a gap that is widening between water have and have-not regions.

We agree and thank the reviewer for highlighting this erroneous logic. We have removed any references to water “have” and “have not” regions of the world and now simply refer to trends in freshwater storage in the context of existing freshwater stress explicitly.

Line 4-- (in abstract):

“We find basins with existing freshwater stress are drying (losing storage) disproportionately, exacerbating the challenges facing the water stressed versus non-stressed basins of the world.”

Line 102--:

“While previous work has shown that the world’s dry regions are becoming drier while the wet regions are becoming wetter²⁶, this work reveals that the stressed regions of the world are becoming drier while the non-stressed regions of the world have no clear overall trend in freshwater storage.”

--social water stress index (Line 45)

The social water stress index is an existing index which we refer to in-passing (only on lines 46-47) as an example of a multi-dimensional adaptation of a traditional water stress assessment. We do not use this index in our analysis but believe it is a notable concept in the framing of our work. Thus, we believe this passing reference is the appropriate involvement of the term and thus it is not included in the terminology box (Box 1).

--water poverty index (Line 45)

We provide the same justification for the water poverty index as for the social water stress index (see above).

--freshwater storage (Line 57); “Trends in freshwater storage, conversely, represent the evolution of total storage within each basin, defined as the vertical sum of groundwater, soil moisture, surface water, and snow water equivalent storages. Freshwater stress and storage are linked, as freshwater storage becomes a required source of water during periods where demands exceed supply” (Line 72); “Freshwater storage loss is linked to increased drought frequency” (Line 314). How much freshwater storage loss must occur (what volume over what time period) for it to be classified as a loss or stress? Is water loss past, present, or future, because “loss” is used in all three ways throughout the manuscript.

Trends in freshwater storage are central to our analysis, and as such, “freshwater storage trends” is included in our key terminology box (Box 1). The freshwater storage trends we incorporate in our analysis are based on observations from the GRACE satellite mission over the April 2002-March 2016 time period (as we document in Supplementary Table 1). In Figure 1, for categorical plotting purposes only, $\pm 3 \text{ mm yr}^{-1}$ is used as the threshold denoting a clear directional storage trend based on the error level of the underlying observations (Vishwakarma et al, 2018). However, this threshold is not used in the derivation of hotspot

basins, which instead is based on the ratio of freshwater storage trends per basin to $0.4 \times$ mean annual streamflow per basin, as we document in the Methods.

Box 1 ('Basin freshwater status' definition); beginning on Line 69:

“Freshwater storage trends: Year-over-year trends in total freshwater storage. Total freshwater storage is a vertically aggregated measure of water storage that includes groundwater, soil water, surface water, canopy water, and ice and snow water equivalents where present. For simplicity, we refer to negative freshwater storage trends as **drying trends or storage loss** and positive trends as **wetting trends or storage gain.**”

References:

Vishwakarma, B.D.; Devaraju, B.; Sneeuw, N. What Is the Spatial Resolution of GRACE Satellite Products for Hydrology? *Remote Sens.* 2018, **10**, 852.
<https://doi.org/10.3390/rs10060852>

--**social sensitivity to freshwater availability** (Line 58); “To consider social adaptability, we utilized the inverted adaptive capacity dataset” (Line 192)

As we remove all instances of “freshwater availability” the above term is no longer found in the manuscript.

--**ecological sensitivity to freshwater availability** (Line 58); “we derived an indicator from data products from two global ecohydrological studies that assess broad ecosystem sensitivity to freshwater availability variations or perturbations” (Line 190)

See comment above.

--**hotspot basins** (Line 66); “the aim of our hotspot mapping is to ‘minimize the social and ecological impacts of freshwater stress and storage loss given available resources’ by asking ‘where are basins with sensitive ecosystems and limited social adaptive capacity exposed to the combined threats of freshwater stress and storage loss?’” (Line 184; should this be moved up to the top of the manuscript before any results are given?); “To classify the global gradient in social-ecological vulnerability into hotspot basins, we implemented a classification approach developed for heavy-tailed distributions which was applicable to the global social-ecological vulnerability distribution” (Line 193); “social-ecological vulnerability hotspots to limited freshwater availability” (Line 210); “hotspots (i.e. basins with ‘high’ and ‘very high’ vulnerability)” (Line 214); “hotspot basins identify the basins with sensitive ecosystems and adaptability-limited societies exposed to the combined threats of freshwater stress and storage loss” (Line 237); “The inverse function curves represents curves of equal vulnerability, and differentiate basins into discrete vulnerability classes used for hotspot identification” (Line 258)

Why use the term “hotspot basin”? Define “hotspot”. “Spot” and “basin” seem at odds because a basin is by definition the largest hydrological unit (a basin is larger than a watershed) so it is hardly a spot; see Supplementary Material Line 171 (“we apply a coarse basin discretization scheme”). Authors might be referencing McClain et al. 2003 (“Biogeochemical Hot Spots and

Hot Moments...”), but in that paper, “hotspots” are defined as “areas (or patches) that show disproportionately high reaction rates relative to the surrounding area (or matrix).” A basin has no surrounding area, as it encompasses all of the area of the hydrological unit.

We thank the reviewer for this reflection on using the term ‘hotspot basin’ and we agree that there are many potential definitions that readers may associate with the term. We did not base our use of the term on McClain et al. 2003; rather we were motivated, in part, by Myers et al. 2000 (“Biodiversity hotspots for conservation priorities”). In Myers et al., hotspots are “areas featuring exceptional concentrations of endemic species and are experiencing exceptional loss of habitat” and these hotspots cover broad geographic areas. Generically, this hotspot definition creates a template of: “areas featuring exceptional [properties pertinent for conservation prioritization]” which are independent of an individual area’s properties relative to surrounding areas. However, in full agreement that this is only one interpretation of the term “hotspot” we have added “hotspot basin” to our key terminology box (Box 1) to make our use of the term explicit. We also understand the reviewer’s inclination that “spot” and “basin” appear at odds, and this is true under certain definitions of the word spot. However, there are multiple instances of large geographic areas being identified as “hotspots” (from Myers et al, as referenced above, to Rodell et al 2018. - a global hydrology paper). Our preference is thus to keep the term “hotspot basin” to differentiate this work from existing studies while strengthening its association with the key publications we list above.

Box 1 (‘Hotspot basin’ definition); beginning on Line 69:

*“**Hotspot basin:** Highlighted basins that possess the greatest vulnerability scores. We identify hotspot basins to support their prioritization in global water resources and integrated management initiatives. Basins are considered hotspots if sorted into ‘high’ and ‘very high’ vulnerability classes following a categorical classification of the numerical vulnerability results.”*

References:

McClain, M., Boyer, E., Dent, C. et al. Biogeochemical Hot Spots and Hot Moments at the Interface of Terrestrial and Aquatic Ecosystems. *Ecosystems* **6**, 301–312 (2003). <https://doi.org/10.1007/s10021-003-0161-9>

Myers, N., Mittermeier, R., Mittermeier, C. et al. Biodiversity hotspots for conservation priorities. *Nature* **403**, 853–858 (2000). <https://doi.org/10.1038/35002501>

Rodell, M., Famiglietti, J.S., Wiese, D.N. et al. Emerging trends in global freshwater availability. *Nature* **557**, 651–659 (2018). <https://doi.org/10.1038/s41586-018-0123-1>

With so many indicators of vulnerability and limitations, it is unclear what the hotspot refers to. Are the hotspots the basins which are unions of being stressed, having decreased storage trend, having low social adaptability, and having high ecological vulnerability? Or just one or a smaller combination of those? There are different categories of hotspots (e.g., ecological vs. social hotspot; Line 201)? Figure 3 shows ecological hotspots and social hotspots separately (Line 260)? Consider saying ecological-only or social-only hotspots to differentiate from

“hotspot”, which refers to a specific subset of general hotspots. Do hotspots include “very high” and “high” vulnerability? This should be clarified at the first mention of “hotspot”.

The reviewer is correct in identifying that there are a number of possible configurations of basin freshwater status, social adaptability, and ecological sensitivity that can lead to a basin being classified as a hotspot basin. However, as the global vulnerability results reveal a heavy-tailed distribution, and as our classification approach limits hotspot basin membership to the ~14% most-vulnerable basins globally (which possess elevated vulnerability scores that are only attainable if basins score considerably high across all vulnerability components), the vast majority of hotspot basins suffer/possess all conditions described by the reviewer.

Reflecting on our presentation of hotspot basin results (through responding to this comment), we found Figure 3 in our original submission counterproductive by providing the ecological-only vulnerability and social-only vulnerability results in the same format as the hotspot basins. To remove this potential confusion, we have removed ecological-only hotspot and social-only hotspot results but preserve the presentation of the underlying ecological and social vulnerability results in Figure 3e,f.

Thus, in the revised manuscript, we now present hotspot basin results only for the combined case of social-ecological vulnerability (i.e. Figure 3a) and we clarify that the hotspot basins are those classified with “high” and “very high” vulnerability scores in the key terminology box, main text (see quotation below), and Figure 3.

Line 217--:

“These hotspot basins consist of basins receiving ‘high’ and ‘very high’ vulnerability scores through our classification procedure.”

--drying trends (Line 81); “losing freshwater storage (drying)” (Line 86)

We have added “drying trends” to our key terminology box (Box 1). See above comment for “--freshwater storage”.

--wetting trends (Line 82); “gaining freshwater storage (wetting)” (Line 89)

We have added “wetting trends” to our key terminology box (Box 1). See above comment for “--freshwater storage”.

--social vulnerability (Line 127); “product of inverted social adaptability and the combined freshwater stress indicator represents social vulnerability” (Line 257)

We have included “vulnerability” in our key terminology box (Box 1), and note that readers can find descriptions of the social and ecological sub-components of vulnerability in the text and Methods. See comment for “--vulnerability”, below.

--social adaptability (Line 128); “the ability of the system to respond to disturbances” and is derived based on input indicators of governance, economic strength, and human development” (Line 130); “a proxy for social ability to respond and adapt to freshwater stresses” (Line 340)

As this term is explicitly defined in the text (see quotation below), we do not add it to the terminology box. Additionally, the use of this term and its associated data product is well referenced to their original source (Varis et al., 2019). On this basis, we do not think that further elaboration is necessary as critical aspects of social adaptability are discussed at length in Supplementary Information Section 1.

Line 139--:

“Social adaptive capacity (Fig. 2a), or adaptability, represents “the ability of the system to respond to disturbances”²⁹ and is derived based on input indicators of governance, economic strength, and human development.”

Supplementary Information Line 99--:

“Our representation of social adaptive capacity was derived by Varis et al.⁴, who understood social adaptive capacity to represent the broad ability of the “social part of social-ecological systems” to both reactively and proactively adapt and increase resilience to ecological vulnerabilities. Varis et al. derive adaptive capacity from three input indicators: government effectiveness, strength of economy, and human development. We note that this is a relatively parsimonious conceptualization of adaptive capacity which may be challenged to represent critical dynamic properties of the social system, such as its absorbability or transformability.⁸”

Reference:

Varis, O., Taka, M. & Kummu, M. The Planet’s Stressed River Basins: Too Much Pressure or Too Little Adaptive Capacity? *Earth’s Future* **7**, 1118–1135 (2019).

--**threat** (Line 134); "combined threats of freshwater stress and storage loss" (Line 134; isn't "storage loss" = "drying"?)

We have removed all use of the term “threat”, which was used only four times in the original manuscript. One instance of the term threat that we do not remove is found in a quoted definition of biodiversity hotspots from Whittaker et al. 2005 (Line 192).

Reference:

Whittaker, R. J. et al. *Conservation Biogeography: assessment and prospect. Diversity and Distributions* **11**, 3–23 (2005).

--**combined stresses** (Lines 127, 138, 143)? What are these? Is it “drying” in a “stressed basin”? It is confusing when “freshwater stress” is one of the “combined stresses”. I see “combined freshwater stress” is defined further down in Line 147 as “co-occurring freshwater stress and storage loss”. Consider moving this definition up to its first use (Line 127).

There needs to be a different word for co-occurring freshwater stress and storage loss (must not be “freshwater stress” because it is circular; Line 147); even more confusing, “freshwater stress indicator” is defined by the relationship between freshwater stress and storage trends.

We agree with the reviewer that it is both confusing and unhelpful for the co-occurrence of freshwater stress and freshwater storage loss to be referred to as “combined freshwater stress” or the “combined freshwater stresses”.

In the revised manuscript, we use the term “basin freshwater status” to refer to the indicator representing the co-occurrence of freshwater stress and freshwater storage loss. We include “basin freshwater status” in our key terminology box (Box 1) - see quotation of its definition, below. This change in terminology should reduce the confusions raised above as there is no longer any mixed use of the terms stress or availability. We have revised the text to replace all use of these terms with “basin freshwater status” or simply co-occurring freshwater stress and storage loss where use of the term “basin freshwater status” is not necessary.

Box 1 (“Basin freshwater status” definition); beginning on Line 69:

*“**Basin freshwater status:** An integrated indicator that combines normalized freshwater stress and normalized freshwater storage trends at the basin scale. High indicator scores are assigned to basins with co-occurring freshwater stress and drying trends. We refer to high freshwater status scores through status **severity**.”*

--economic inequality (Line 158); is this inequality within or among nations? Move up definition from Line 162?

Economic activity, like all dimensions incorporated in this analysis, is evaluated at the basin scale and these basins exist both within and among nations. We do not provide a definition of economic inequality on line 162 and thus are unable to address this comment.

--vulnerability (Line 188); “the product of (i) ecological sensitivity, (ii) inverted social adaptive capacity, and (iii) the combined freshwater stress indicator” (Line 188)

A definition of vulnerability is now included in the key terminology box (Box 1).

Box 1 (“Vulnerability” definition); beginning on Line 69:

*“**Vulnerability:** The likelihood of society and ecosystems to experience harms due to exposure to freshwater stress and storage loss when considered together as a basin’s freshwater status. This vulnerability definition is an application of Turner et al.’s generic definition. Vulnerability is quantified using social adaptability, ecological sensitivity, and basin freshwater status indicators. Social adaptability and ecological sensitivity indicators are described in the text and Methods.”*

--ecological vulnerability (Line 257); “product of ecological sensitivity and the combined freshwater stress indicator represents ecological vulnerability” (Line 256)

See response to “--social vulnerability”, above.

--social-ecological system vulnerability (Line 382); “We analyze social-ecological system vulnerability as the product of (i) exposure, (ii) ecological sensitivity, and (iii) social adaptive capacity all with respect to freshwater stress and freshwater storage trends. Sensitivity and adaptive capacity are often considered separately in vulnerability assessments however the two concepts are highly interlinked, and we consider the two together as it supports our focus on

evaluating the coupled social and ecological vulnerability of basins.” (Line 382); “We derive social-ecological vulnerability to freshwater stress and storage loss as the product of (i) exposure to freshwater stress and storage loss and (ii) the combination of inverted social adaptive capacity and ecological sensitivity to hydrological disturbance, both of which are represented by derived indicators”; “Social-ecological vulnerability is thus represented in our analysis through equation (1)...” (Line 415)

See response to “--vulnerability”, above.

--resilience (Line 388); “resilience, which can be illustrated by the feedback effects of social and ecological impacts of freshwater stress and storage trends that may alter social and ecosystem functioning and/or well-being” (Line 388)

Resilience is a key concept in social-ecological system analysis and is closely linked to the concept of vulnerability. On this basis, we believe that highlighting this linkage is important, as we do in the Supplementary Information (see quotations below), but do not believe it requires inclusion in the key terminology box.

Supplementary Information *Line 122--*:

“Resilience is considered conceptually through the feedback mechanisms shown in the conceptual model that act on the underlying process-based controls of ecosystem and social functions. Ecological impacts ... reduce ecosystem resilience by decreasing system diversity, connectivity, and altering ‘slow variables’ that regulate ecosystem functioning¹⁰. ... Social impacts of freshwater stress and storage loss can feed back and deteriorate process-based controls of social resilience, though we note that literature is sparse on cascading impacts of hydrological hazards in large-scale social-ecological systems.”

--exposure indicator (Line 399); “the spatial co-occurrence and severity of freshwater stress and trends in freshwater storage” (Line 399)

In the revised manuscript, we no longer find any instances of the term “exposure indicator”. However, we do define vulnerability as the product of exposure to basin freshwater status and social-ecological sensitivity. We provide clarification on our use of the term exposure in the Methods:

Line 398--:

“We analyze social-ecological system vulnerability as the product of (i) exposure to freshwater stress and storage loss, represented by the basin freshwater status indicator, and (ii) the combination of ecological sensitivity and social adaptive capacity, which we represent using derived indicators.

Basin freshwater status is derived to represent exposure to the spatial co-occurrence and severity of freshwater stress and freshwater storage trends. This approach enables trends in freshwater storage to differentiate basins of equal freshwater stress as storage trends can aggravate or offset existing stress levels depending on the direction of the storage trend and the existing stress level.”

--social-ecological sensitivity indicator (Line 404); “the ecological ability to either adapt or absorb freshwater availability perturbations, and social ability to adapt proactively and reactively to generic stresses” (Line 404)

Sensitivity is contextualized with respect to vulnerability in the key terminology box. See comment for “--vulnerability”, above.

-Some terms are defined by precise thresholds (freshwater stress is defined by a ratio of withdrawal/streamflow > 0.1) and equations (social-ecological system vulnerability is defined by Equation 1; Line 415), while others are not. These thresholds and equations should be made clear earlier, possibly at first mention of the term. Does “drying” have a threshold (perhaps a given percentage of water volume over a given period of time)?

See comment provided for “--drying trends”, above. Where percentile-based thresholds are used to describe results, these threshold values are shown in all figure legends.

-Describe time windows of all “trends”; are they past, past-up-to-present, or anticipated future trends?

The temporal range of all data are documented in Supplementary Table 1.

-What is the global coverage of basins in this study? How many basins (by count and area) were and were not represented in this study relative to the total basins in the world (provide ratios); Line 375?

We have included the number of basins represented in the study in the main text (see quotation below) and provide the percentage of basins of the global set included in our analysis by count and area in the Methods (see quotation below).

Line 65--:

“In this study, all analyses are performed at a large basin scale ($n = 1204$, median area $\sim 70,000$ km²) ... ”

Line 385--:

“We masked basins in Greenland, northern Canada, and several small islands from our analysis due to inconsistent data coverage. This masking reduced the HydroBASINS level 4 discretization scheme to 1204 basins from an original set of 1341 (90% retention by count, 97% by surface area).”

-What are the actual impacts of limited freshwater availability? Why does analyzing indices of freshwater availability matter? Please elaborate more and in a concrete way.

We have a dedicated discussion on the implications for hotspot basins (lines 315-335), where we detail the many impacts of freshwater stress and storage loss. These include: environmental flow transgression, land-surface energy balance impacts, access inequalities due to wells running dry, and so on. Thus, we are admittedly somewhat confused by the first question “what are the actual impacts of limited freshwater availability?” Further, a core message of our analysis is that studying freshwater stress and storage loss in isolation is insufficient for a robust vulnerability analysis and, rather, they need to be complemented with

considerations of social and ecological sensitivity to these phenomena. On this basis, analyzing indices of freshwater becomes necessary to enable integration with other considerations that are non-hydrological (i.e. human and ecological variables). Thus, their importance (i.e. the degree to which these indices “matter”) is tied to the question of how important vulnerability analysis of this nature is itself. Given that both of our freshwater indicators (freshwater stress and storage trends) are widely used across the hydrological literature, a defense of their use may seem out of place in this manuscript.

-The authors discuss social-ecological systems throughout the manuscript, but they present figures delineated only by hydrological system boundaries. Given that this manuscript centers largely around maps (all figures are world maps) and the authors chose to use this uncommon delineation of space by hydrological basin boundaries rather than by national borders or ecosystem boundaries, the authors need to be clearer about how they converted national and ecological statistics into basin statistics. If figures show basin boundaries that cross national boundaries, then how is, for example, social adaptability calculated within a hydrological basin boundary and also across a national boundary? Some of the answer is mentioned in the METHODS but it should be made clearer in the main text.

We agree with the reviewer that basins are an unconventional delineation of space. However, basins are an increasingly used template for hydrologically-based social-ecological systems analysis (e.g. see Varis et al. 2019 who propose river basins as a suitable unit of analysis for social-ecological systems). Of course, basins are also a common unit of analysis in the hydrological sciences and will be met without surprise by this work’s hydrology audience. We have added a sentence to the main text that comments on our use of basins as the underlying spatial template:

Line 64--:

“Basins, at various scales, are an increasingly used and particularly suitable geospatial unit of analysis for hydrologically-based social-ecological systems analysis.”²³

Reference:

Varis, O., Taka, M., & Kummu, M. (2019). The planet's stressed river basins: Too much pressure or too little adaptive capacity?. *Earth's Future*, 7: 1118– 1135.
<https://doi.org/10.1029/2019EF001239>

All of our input data (with one exception, described below) are provided at resolutions finer than national and basin scales (in fact - most are continuous raster data, meaning data are provided at the level of individual grid cells with original grid cell resolutions specified in Supplementary Table 1). Thus, our approach to summarize data to the basin scale follows standard procedures: either taking the area-weighted average of within-basin grid cells or the sum of all within-basin grid cells, depending on the nature of the data being summarized (i.e. if the data represents intensive vs. extensive properties). As these procedures are standard across “lumped” spatial analyses, we believe that documenting them within the Methods is sufficient. However, we have added a reference to the Methods when introducing our use of basins in this study, and provide greater detail in the geospatial methods section:

Line 65--:

“In this study, all analyses are performed at a large basin scale ($n = 1204$, median area $\sim 70,000 \text{ km}^2$) and input data align to the year 2015 as best as possible (see Methods for data preprocessing and geospatial method details).”

Line 382--:

“All data were summarized to the basin scale by (i) calculating the area-weighted average value of intensive properties within each basin using an algorithm to calculate cell area on a geographic grid⁵⁴, or (ii) calculating the within-basin sum of extensive properties.”

For more information on the application of basins as our spatial unit of analysis, see Supplementary Figures 4-6 (maps revealing the summarizing of individual datasets to the basin scale). Supplementary Table 1 describes all data processing steps, while Supplementary Figure 3 describes the workflow of our analysis (i.e. how data is handled throughout the hotspot basin derivation process).

The exception, alluded to above, that is not provided at the grid cell level (i.e. non-continuous raster data) is national-level IWRM implementation data. Notably, this data isn't used in the derivation of hotspot basins but rather is used for discussion by comparing these data with the hotspot basin results. These national-level IWRM data are summarized to the basin scale by taking an area-weighted average of IWRM data within each basin. Effectively, this means that if a basin is completely within a single nation it will receive the exact IWRM score of that nation. However, if it is a transboundary basin, it will receive an IWRM score that is the area-weighted average of all IWRM scores within the basin.

Line items:

Line 21: Change “enable” to “enables”; delete “greater”

Done.

Line 68: This section is titled “The global co-occurrence of freshwater stress and water availability trends”, but (fresh) water availability was just earlier defined as “the global co-occurrence of freshwater stress and freshwater storage trends” (Line 62), so “freshwater stress” is redundant (i.e., it is named explicitly and then again implicitly in “water availability”). Why not title this section “The global freshwater availability trends” or “The global co-occurrence of freshwater stress and freshwater storage trends”?

We have changed as suggested to: **“The global co-occurrence of freshwater stress and freshwater storage trends”** (now Line 72).

Line 76: Change “where” to “when”

Done.

Line 96: “by-and-large” is too informal

Removed.

Line 115: The figure caption is titled “Global co-occurrence of freshwater stress and trends in storage”, so again there is a problem similar to in Line 68. Why not simplify to “Global freshwater availability trends”?

Given that we have altered the section title (previously Line 68, see response above), our slightly modified figure caption is now consistent with this revision: “Fig. 1: Global co-occurrence of freshwater stress and storage trends”.

Line 246: 98 or 97 %?

We have restructured this sentence to avoid this potential confusion - see below.

Line 251--:

“Performing 10,000 realizations for each uncertainty analysis, we find that 98% of the identified transitional and hotspot basins are identified as at least transitional basins in over 50% the realizations considering spatially uniform uncertainty, and 97% when considering spatially variable uncertainty (Supplementary Figures 8,9).”

Line 259: “represents” should be “represent”

Fixed.

Line 285: Move “(input data align to ~2015)” up earlier to where input data are first described

Done - “input data align to the year 2015” now appears on Line 67.

Line 303: There are basins in the figure that look gray and shaded (e.g., most of North America; India). This is confusing because in Fig 4a, the caption says “Gray basins are hotspot basins with no data on IWRM implementation levels” but in the scale bar it says “Non-hotspots are shaded to emphasize hotspots”.

We have revised the labelling of this figure for clarity. Basins are ‘darkened’ if they are not hotspot basins (i.e. so that hotspot basins appear ‘brighter’) -- as IWRM data is not used in the derivation of hotspot basins it is possible for a basin to be a hotspot but not have IWRM implementation data. We also note that we have updated the IWRM data included in this figure (additional data were released between the submission of the original manuscript and this revision), and now USA and India (as noted above) have IWRM data.

Line 308--:

“Figure 4: Integrated water resources management in hotspot basins. a, Map of IWRM implementation levels overlaid by hotspot basin results. Non-hotspot basins are shaded dark grey to emphasize hotspot basins (no shading) ... ”

Line 315: Is this a tautology? Generally, resilience is defined as success in the face of hardship/limitation, so here resilience should be agnostic to the drying conditions. Drying conditions should not decrease ecosystem resilience to drought. See definition in

Supplementary Material Line 30 and 37. There, resilience is defined outside of the perturbation (e.g., drying conditions).

While we agree with the reviewer that resilience can be simply described as “success in the face of hardship”, a missing element of this perspective is that resilience is a dynamic system property that can change through time. Thus, “hardships/limitations” (such as freshwater storage loss) can affect system resilience in future states through feedback mechanisms. If resilience were not a dynamic property, efforts to promote system resilience would not exist; however this is a major focus within social-ecological systems research. Furthermore, as the text in question directly cites a paper that explicitly uses this resilience language (Fan et al. 2017), we would prefer to keep the text as-is.

Reference:

Fan, Y., Miguez-Macho, G., Jobbágy, E. G., Jackson, R. B. & Otero-Casal, C. Hydrologic regulation of plant rooting depth. *PNAS* **114** 10572–10577 (2017).
<https://www.pnas.org/content/114/40/10572>

Line 324: Conflict is pervasive and inevitable, so its triggering is not that meaningful. Is there persistent or violent conflict linked to freshwater stress? If so, that would be worth mentioning.

We agree with the reviewer that connecting freshwater stress and storage loss to persistent conflict (in contrast to linking only with instances of conflict) is a critical distinction and would be an important contribution. However, the water conflict database we rely on does not provide conflict duration or reoccurrence data. In acknowledgement of this comment, however, we have made a text modification to include the perspective of considering the role of freshwater stress and storage loss in relation to **sustaining** conflict (see quotation below).

Line 332--:

“As this analysis considers only a subset of water related stresses, this serves to indicate the potentially leading role freshwater stress and storage loss may play in instigating, contextualizing, and sustaining social conflict.”

Line 346: Wording? How about “whose data currently exist delineated by national jurisdiction rather than basin boundaries”?

Changed to (now line 352): “... which does not yet have data available globally at a sub-national scale”.

Line 358: “analysis” should be “analyses”

Fixed.

Supplementary Material

Line items:

Line 42: There are three grey boxes and two shades of grey. It is “grey” here but “gray” in Fig. 4.

We have altered the figure and legend to remove this confusion.

Line 94: Add “to” after “(increases)”

This line is removed in our pared down Supplementary Information.

Line 145: “feedback” is two words

It can also be written as one word:

<https://dictionary.cambridge.org/dictionary/english/feedback>

Line 147: How does climate change result in increasing rates of freshwater demand? Is it more plausible that climate change results in decreasing freshwater supply while population increase results in increasing freshwater demand? Also, climate change (as a significant driver of loss of freshwater storage) is not discussed in the main body of the manuscript, so either discuss climate change more in the main body, or less in the Supplementary Material.

There are multiple mechanisms through which climate change can result in increasing rates of freshwater demand (higher evaporative demand, longer growing seasons, less green water leading to greater need for irrigation, larger cooling demand for energy use, etc.). Rodell et al. (2018) discuss climate change as a major driver of trends in freshwater storage. In our conceptual model, we represent the impact of climate change through dashed lines, indicating we do not consider climate change explicitly in our analysis; however we raise these potential mechanisms in the Supplementary Information to highlight limitations of our approach and potential avenues for future work. Furthermore, we only explicitly discuss climate change once in the Supplementary Information -- thus discussing climate change less in the Supplementary Information would require removing altogether, which we perceive as unnecessary.

References:

Rodell, M., Famiglietti, J.S., Wiese, D.N. et al. Emerging trends in global freshwater availability. *Nature* **557**, 651–659 (2018). <https://doi.org/10.1038/s41586-018-0123-1>

Table 1: Crop calorie production. The temporal range of this data set is 1997–2003 but the target year of analysis is 2015. This may result in a serious underreporting of crop production. The world population in 2003 was 6.4B, whereas the world population in 2015 was 7.4B (an increase of 16%). Can you scale your crop calorie production to reflect this increase in calorie demand?

In the revised manuscript, we have replaced the Cassidy et al. (2013) crop production dataset with a more recent dataset produced by Kummu et al. 2021 that corresponds to global food crop production for the year 2010. This dataset is based on a yield to kilocalorie conversion for the 27 food crops within the SPAM dataset (<https://www.mapspam.info/>). While this new dataset still does not align with the target year of 2015, it is considerably closer. We have decided not to scale this data to the year 2015 as we predominantly discuss the crop production summary statistics in global percentages and do not believe that linearly scaling to the year 2015 will make a substantial difference in the percentage results to a degree that would merit the additional methodology and justification. We thank the reviewer for raising this concern and believe that the replacement food crop production dataset that we have implemented improves the relevance of our discussions on crop production substantially.

References:

Kummu, M., Heino, M., Taka, M., Varis, O. & Viviroli, D. Climate change risks pushing one-third of global food production outside the safe climatic space. *One Earth* 4, 720–729 (2021).
<https://www.sciencedirect.com/science/article/pii/S2590332221002360>

Line 220: “Bain” should be “Basin

Fixed.

Reviewer 2

I was VERY excited when I started reading this paper because of the multidimensional approach to water scarcity and social vulnerability and also because it incorporated remotely sensed hydrology. I think the approach is clever and innovative.

I was actually quite surprised (and a little disappointed) when the results of this clever, multidimensional approach highlighted essentially the same regions as previous studies (over the last couple of decades) have found using the much simpler metrics (like demand/supply). But of course, this makes sense because all this vulnerability is likely due to human water demand and overuse of water resources. However, the additional information about how water scarcity, storage trends and social vulnerability indicators overlap with population, agriculture, economic activity and biodiversity adds substantially to that story and is an important finding.

We thank the reviewer for these reflections. To add to them, we believe an important contribution of this analysis is the additional context it provides to regions that have previously been highlighted using simpler metrics. Indeed, our methodology lends itself to highlight similar regions as previous studies as our hotspot basins are constrained to basins experiencing freshwater stress (essentially a demand/supply metric) and freshwater storage loss. However, co-occurrence between freshwater stress and freshwater storage loss is not a necessary condition and thus our finding that freshwater storage loss is occurring in many stressed basins further contextualizes the challenges facing these basins. Furthermore, we believe that focusing on the differential vulnerability of these stressed and drying basins is useful for prioritization considerations in global sustainable development initiatives, among others.

Where this paper becomes truly IMPORTANT is when the authors evaluate IWRM efforts across the globe and assess how IWRM efforts overlap with hotspot basins. This is BRILLIANT and an incredibly important finding. This paper should be both motivational and useful to global efforts to advance IWRM.

We thank the reviewer for their supportive review of our manuscript.

The paper is innovative, well written and important; it should be published. I only have one tiny remark: the statement in lines 179 through 181 is really not true. Vörösmarty and collaborators, and other groups, have published many papers that identify hotspots of water scarcity and water insecurity issues. I would delete, or at least modify, this statement.

We have modified this sentence significantly (see below for updated text) and apologize for any unintended misrepresentation in the original manuscript.

Line 187--:

“Hotspot mapping has been a successful endeavor within the field of conservation biogeography^{31,32}, and many global hydrology studies have identified regions of exceptional water scarcity and security challenges.^{e.g.13–15,17–19”}

Reviewers' Comments:

Reviewer #1:

Remarks to the Author:

See attached pdf.

Reviewer Report for *Hotspots of social and ecological impacts from freshwater stress and storage loss*

Kudos to the authors for their thorough revision and response. They did a good job clarifying their terms—the addition of Box 1 is a great idea—which communicates their ideas much better. However, I think the manuscript still suffers from a few remaining major issues (and some line items) that must be addressed before it is ready for publication. I made a few general comments at the end.

Remaining major issues (blue text is from the authors in their Rebuttal)

1. The maps/analyses were done very well, and the main body should flow well enough to describe and highlight them. However, in the manuscript (and also defended by the authors in their Rebuttal), there is a pattern of placing some descriptions of methods only in the Methods or Supplementary Information sections, or referring to other papers for specific use of terms and indices. Surely, that is fine and appropriate, but there also needs to be some high-level, general description of the methods in the main body of the text first (for example, your use of Box 1 is nice because it brings all relevant terminology to the top of the manuscript, provides clear definitions, then refers the reader to the Methods for further information). Otherwise, the story is fragmented, incomplete, and difficult to follow. Most readers will not look up all of these additional materials to fact-check the story in the main body.

- *Box 1 (within ‘Freshwater storage trends’ definition); beginning on Line 69: “For simplicity, we refer to negative freshwater storage trends as **drying trends or storage loss** and positive trends as **wetting trends or storage gain**.”*

A loss/gain can occur instantaneously. A trend is the integration of competing losses and gains over a period of time longer than an instant, but if that period of time is not defined, then this appears to be ambiguous. Perhaps you can say “net loss/gain”?

Generally, I think your use of “trend” is too loose. Why give the time period only in the Supplementary Information when you can give it in the main body parenthetically after the mention of “trend”? You describe the data in Line 290 “(input data align to ~2015)” so I think you can describe the trend similarly.

The freshwater storage trends we incorporate in our analysis are based on observations from the GRACE satellite mission over the April 2002-March 2016 time period (as we document in Supplementary Table 1). In Figure 1, for categorical plotting purposes only, $\pm 3 \text{ mm yr}^{-1}$ is used as the threshold denoting a clear directional storage trend based on the error level of the underlying observations (Vishwakarma et al, 2018). However, this threshold is not used in the derivation of hotspot basins, which instead is based on the ratio of freshwater storage trends per basin to $0.4 \times$ mean annual streamflow per basin, as we document in the Methods.

This is so beautifully clear. It is exactly what is needed in the definition of the trend. I think it would be completely appropriate to include a truncated version in the main body: “freshwater storage trends (2002–2016; specific thresholds outlined in Methods and Supplementary Table 1).”

“Freshwater storage trends: Year-over-year trends in total freshwater storage.

Sounds like a tautology.

taking the area-weighted average of within-basin grid cells or the sum of all within-basin grid cells, depending on the nature of the data being summarized

Why not add this to the main body instead of Methods? I think this is crucial to telling your story and it does not require a lot of words. The Methods and Supplementary Information are where you can expand on this, but this is still required in the main body because otherwise the main body is hard to follow.

2. There are many sentences that are too long and run on (e.g., Line 1, Line 30, and others listed below). Some sentences have repeated words that sound awkward (Line 170: “which aims to “reduce the number of people suffering from water scarcity”, which...”; Line 273: “tracks IWRM implementation at the national scale to track...”; Line 285: “levels across all levels...”).

3. The setup of the social aspect of this work needs to be clearer. After reading this revision of the manuscript, it seems that hotspots refer more to vulnerability to future problems, rather than impacts from past problems. Is this the intention? Is this analysis really showing hotspots of social and ecological “impacts from” freshwater stress (as in the title) or, rather, “vulnerability to” it? I’m confused because the social aspect in this analysis is measured specifically as the (“current” [i.e., 2015 in the Varis et al data set]) capacity of a society to adapt to future problems, which is then mapped onto a society’s vulnerability to the future problem of freshwater stress. Also, you repeatedly mention the potential for (future) impacts (Line 224 and 395), rather than actual (past) impacts. Thus, the title doesn’t seem to match the main body.

Line items

Line 1: Is this a run-on sentence? Consider separating “cycle however” with a period or semicolon.

Line 27: “global freshwater crisis” seems jarring because so far, you say only how people “influence”, “stress”, and even “dominate” the hydrological cycle, but that doesn’t necessarily lead to “crisis”. Below, on Line 161, you state the positive impacts of “stress”. Consider defining the crisis specifically (eg, lack of freshwater quantity, poor water quality, death?) or simply adding a reference.

Line 30: Consider rewording. Currently, it reads as “we consider impacts of xx and xx by synthesizing a subset of xx and xx datasets that exist with xx, xx, and xx datasets.”

Line 54: Change “behvaiour” to “behaviour”.

Line 55: Consider adding a comma between “research and.”

Line 57: “these three fields” Which three? The sentence is long so I stopped counting the fields. Are they water scarcity, water security, and social-ecological systems research? It is unclear, given the way the previous sentence is written (the combination of social-ecological makes it seem like there can be four). Perhaps add a comma (see comment for Line 55) and/or add “additional” between “integrating concepts”.

Line 60: Change “tress” to “stress”.

Line 62: Consider that the total list (ie, global gradient) and the list of hotspots-only are separate objectives. They are products that are useful in different ways. Otherwise, as it is, this sentence is too long and could be broken up with a comma before “and identify”.

Box 1 Freshwater stress: The second sentence (“The W/Q ratio...”) just restates the first sentence (“The ratio of...”). You can cut out the second sentence, keeping the first, and replace “evaluated” with “calculated”. Also, how about not using stress twice, so “We refer to basins with $W/Q \geq 10\%$ as **stressed basins**...”?

Box 1 Freshwater storage trends: Please add the years of the trends here parenthetically and with reference to the Methods.

Line 73: “We mapped freshwater stress and trends in freshwater storage” How about something like “We mapped current freshwater stress and past-to-current (c. 2002–2016) trends in freshwater storage” or something similar? It doesn’t really matter how you word it as long as you distinguish that the stress is a *present state of things*, whereas storage is a recently past trend over a period of multiple years. You can do this in Box 1 first, so you don’t have to change the subsequent text?

Line 85: Consider adding a comma between “stress as” and “stress while”.

Line 94: The term “human activity” (Line 93) is sufficiently vague and opaque that it has no meaning. What specifically and literally does it mean? Typically, I would think of resource extraction, water re-routing, etc, and there are plenty of papers in the literature that you can reference. These specific things don’t need to be included in any analysis, but simply provided as background so the reader can understand why this analysis matters. Can you give one or two specific examples in parentheses, or at least put a reference (similar to how you put a reference for “natural variability” at the end of the sentence)?

Line 103: Great point.

Line 115: This is an important point, but I think it gets lost because the sentence is so long. Consider breaking it into smaller sentences.

Line 116: Consider adding a comma between “basins and”.

Line 121: Consider adding a comma between “distributions as”. Again, this is an important point, but I think it gets lost because the sentence is so long. Consider breaking it into smaller sentences.

Line 133: Consider adding a comma between “coordinate and”, and adding a semi-colon or period between “coordinate with the” while cutting out “with”. The sentence is too long.

Line 146: It would be appropriate to mention in a few words that you lumped social indicators by basin, then refer the reader to the Methods for further information. It is an abrupt transition from the hydrologic status of a basin, to the social (-adaptive) status of a (hydrologic) basin.

Line 158: Consider adding a comma between “development which”. It’s a very long sentence.

Line 170: Consider breaking up this sentence (two “which” seems a bit much).

Line 181: Capitalize “population”? Or change preceding period to comma?

Line 186: Perhaps, to drive the point home about what a hotspot is, consider changing this sentence to “We mapped the global gradient in social-ecological vulnerability to freshwater stress and storage loss at the basin scale and, from this, identified the hotspot basins of vulnerability”. “Basins of vulnerability” seems more in line with what you say (“vulnerability hotspots”) in Line 217. This list of hotspot basins is a top contribution of this paper but it gets a bit lost in the wordiness and repetition of the sentence, which starts with “gradient in social-ecological vulnerability” and ends with nearly the same thing (“global vulnerability distribution”).

Line 244: Change “identify the basins” to “are those”.

Line 247: Start the sentence with “Identification of”?

Line 285: “levels across all levels...” Awkward repetitive phrasing. Consider rewording (also Line 273: “tracks IWRM implementation at the national scale to track...”; Line 170: “which aims to “reduce the number of people suffering from water scarcity”, which...”).

Line 297: So would you consider the hotspots (Afghanistan, Algeria, Argentina, Egypt, India, Iraq, Kazakhstan, Mexico, Somalia, Ukraine, Uzbekistan, and Yemen) that are vulnerable but are not receiving IWRM as being even hotter hotspots? Should this be mentioned in the abstract? Seems important.

Line 317: Change “between” to “among”.

Line 323: Break up sentence or add commas. It’s too long.

Line 348: What is “ref.50” referring to?

Line 363: Break up sentence. It’s too long.

Supplementary Table 2: You switch between present and past tense multiple times (e.g., “we aggregated” and “we calculate” in the “Additional Preprocessing” of “Freshwater withdrawal...”; “we estimated” in the “Additional Preprocessing” of “Streamflow”). Pick one and be consistent.

General comments

-drying may be more commonly associated with drought than anthropogenic withdrawal. However, we also note that use of the term drying is also widely used in the context of human-driven storage loss

I thought, given that this journal is interdisciplinary and not limited to the context of human-driven storage loss (or even hydrology), using the word “drying” needed a clarification. Your use of it (and justification) is absolutely legitimate, but the reader can get confused. Including it in Box 1 is a great idea.

-we have removed ecological-only hotspot and social-only hotspot results but preserve the presentation of the underlying ecological and social vulnerability results in Figure 3e,f.

These updated figures are beautiful and informative.

-We have removed any references to water “have” and “have not” regions of the world and now simply refer to trends in freshwater storage in the context of existing freshwater stress explicitly.

-We have removed all use of the term “threat”

I think this makes your analysis clearer and more objective.

-Thus, “hardships/limitations” (such as freshwater storage loss) can affect system resilience in future states through feedback mechanisms. If resilience were not a dynamic property, efforts to promote system resilience would not exist; however this is a major focus within social-ecological systems research.

I agree with this. My point was that a system can experience drying to a given tolerance without its resilience being affected. The problem is, as you say, feedback mechanisms can push a system past a tipping point into a “future state”, in which it loses/gains resilience. Is that what is happening here, though? Is the freshwater stress within the tolerance of these environmental systems or not?

Is it pushing these systems past their functional boundaries into new states in which their resilience/function changes? I didn't see any language about future states, tipping points, non-stationarity, altered resilience, or the like.

As I understand it, you are showing areas of freshwater stress and then making further implications about resilience in these areas. So, the use of “resilience” was unclear in the first draft. The language (e.g., Lines 315 and 388 in the first draft) in the first draft sounded like it linked the freshwater stress from this study to resilience generally. Here in the second draft, I think you clarify in Lines 363 and 394 that evaluating the resilience of your study basins is outside the scope of this study.

Supplementary Line 126:

-Line 145: “feedback” is two words

It can also be written as one word:

<https://dictionary.cambridge.org/dictionary/english/feedback>

You were using it as a verb, though (“Ecological impacts can also feedback to affect freshwater stress...”).

<https://dictionary.cambridge.org/dictionary/english/feed-back>

I see you changed it in the revision (“Ecological impacts can also feed back...”; Line 126, Supplementary Information). Also, you have it written correctly in Line 131, Supplementary Information.

The reviewer comments are reproduced verbatim in the grey box below. Our responses to each point are in red text in the embedded white boxes. Corresponding edits made in the revised manuscript are highlighted yellow.

Reviewer Report for *Hotspots of social and ecological impacts from freshwater stress and storage loss*

Kudos to the authors for their thorough revision and response. They did a good job clarifying their terms—the addition of Box 1 is a great idea—which communicates their ideas much better. However, I think the manuscript still suffers from a few remaining major issues (and some line items) that must be addressed before it is ready for publication. I made a few general comments at the end.

We thank the review for their thorough and thoughtful comments on our manuscript. The manuscript has improved significantly because of their attention to detail and insight. We want to acknowledge and voice our appreciation for their influence on this work, which we find exceptional. In our revised manuscript, we:

- Address all remaining major issues raised by the reviewer (i.e. moving key methodological details to the main text from the methods section; and clarifying our setup of the vulnerability component of the work)
- Implement all line item changes suggested by the reviewer

Remaining major issues (blue text is from the authors in their Rebuttal)

1. The maps/analyses were done very well, and the main body should flow well enough to describe and highlight them. However, in the manuscript (and also defended by the authors in their Rebuttal), there is a pattern of placing some descriptions of methods only in the Methods or Supplementary Information sections, or referring to other papers for specific use of terms and indices. Surely, that is fine and appropriate, but there also needs to be some high-level, general description of the methods in the main body of the text first (for example, your use of Box 1 is nice because it brings all relevant terminology to the top of the manuscript, provides clear definitions, then refers the reader to the Methods for further information). Otherwise, the story is fragmented, incomplete, and difficult to follow. Most readers will not look up all of these additional materials to fact-check the story in the main body.

We agree with this suggestion and have modified the main text to include details on (i) the temporal range of the freshwater storage trends, and (ii) data resampling methods. See individual responses to these issues, raised below, for details on each one.

- Box 1 (within 'Freshwater storage trends' definition); beginning on Line 69:
"For simplicity, we refer to negative freshwater storage trends as **drying trends** or **storage loss** and positive trends as **wetting trends** or **storage gain**."

A loss/gain can occur instantaneously. A trend is the integration of competing losses and gains over a period of time longer than an instant, but if that period of time is not defined, then this appears to be ambiguous. Perhaps you can say "net loss/gain"?

See next comment.

Generally, I think your use of "trend" is too loose. Why give the time period only in the Supplementary Information when you can give it in the main body parenthetically after the mention of "trend"? You describe the data in Line 290 "(input data align to ~2015)" so I think you can describe the trend similarly.

The freshwater storage trends we incorporate in our analysis are based on observations from the GRACE satellite mission over the April 2002-March 2016 time period (as we document in Supplementary Table 1). In Figure 1, for categorical plotting purposes only, ± 3 mm yr⁻¹ is used as the threshold denoting a clear directional storage trend based on the error level of the

underlying observations (Vishwakarma et al, 2018). However, this threshold is not used in the derivation of hotspot basins, which instead is based on the ratio of freshwater storage trends per basin to $0.4 \times$ mean annual streamflow per basin, as we document in the Methods.

This is so beautifully clear. It is exactly what is needed in the definition of the trend. I think it would be completely appropriate to include a truncated version in the main body: “freshwater storage trends (2002–2016; specific thresholds outlined in Methods and Supplementary Table 1).”

We have clarified the time period that the freshwater storage trends correspond to in their definition in Box 1. We do not think that using the suggested ‘net trends’ terminology is fully appropriate, however, as the trends are not determined using solely the first and last values in the time series (as may possibly be inferred through the term ‘net’), and rather were determined by Rodell et al. (2018) using a best-fit linear trend of the time series of seasonally-detrended monthly observations over the 2002-2016 time period. We incorporate text very similar to as suggested by the review above in our revised definition of freshwater storage trends in box 1.

Line 71 (inside Box 1):

Freshwater storage trends: Year-over-year trends in total freshwater storage based on satellite observations over the 2002-2016 time period. Total freshwater storage is a vertically aggregated measure of water storage that includes groundwater, soil water, surface water, canopy water, and ice and snow water equivalents where present. For simplicity, we refer to negative freshwater storage trends as **drying trends** or **storage loss** and positive trends as **wetting trends** or **storage gain**.

“**Freshwater storage trends:** Year-over-year trends in total freshwater storage.

Sounds like a tautology.

We agree and think this definition when placed in isolation is redundant in unnecessary. Our revised definition (see above) redresses this, as the definition specifies that the trends are (1) based on **total** water storage, (2) are derived from satellite observations, and (3) correspond to the 2002-2016 time period. I.e. the revised definition provides 3 non-redundant and not-implicit details about the trends we consider.

taking the area-weighted average of within-basin grid cells or the sum of all within-basin grid cells, depending on the nature of the data being summarized

Why not add this to the main body instead of Methods? I think this is crucial to telling your story and it does not require a lot of words. The Methods and Supplementary Information are where you can expand on this, but this is still required in the main body because otherwise the main body is hard to follow.

We agree and have moved this information to the main text. See below.

Line 67--:

In this study, all analyses are performed at a large basin scale ($n = 1204$, median area $\sim 70,000 \text{ km}^2$). Input data align to the year 2015 as best as possible, and data are summarized to the basin scale by computing the area-weighted basin average or within-basin sum, depending on the intensive of extensive nature of each dataset (see Methods and Supplementary Information).

2. There are many sentences that are too long and run on (e.g., Line 1, Line 30, and others listed below). Some sentences have repeated words that sound awkward (Line 170: “which aims to “reduce the number of people suffering from water scarcity”, which...”; Line 273: “tracks IWRM implementation at the national scale to track...”; Line 285: “levels across all levels...”).

We agree and have modified all of the sentences identified by the reviewer accordingly. Our revisions to the sentences highlighted by the reviewer above are provided below, while the other instances of long sentences raised in the ‘Line items’ section are provided there.

Line 1--:

Humans and ecosystems are deeply connected to, and through, the hydrological cycle. However, the impacts of hydrological change on social and ecological systems are infrequently evaluated together at the global scale.

Line 28--:

In this paper, we consider impacts of freshwater stress and storage loss on humans and ecosystems. We do this by synthesizing a subset of the few but critical global ecohydrological and sociohydrological datasets that exist with freshwater storage, freshwater withdrawal, and streamflow datasets (see Supplementary Table 1).

Line 160:

One such example is Sustainable Development Goal (SDG) 6.4 (“reduce the number of people suffering from water scarcity”), which we argue should increasingly be linked to targets of SDG 10 (“reduce inequality within and among countries”).

Line 242:

For IWRM implementation data, we rely on the IWRM Data Portal³⁵ which tracks progress on SDG 6.5.1 (“IWRM implementation at the national scale”).

Line 255:

There is thus a wide range of IWRM implementation across all levels of social-ecological vulnerability to freshwater stress and storage loss, and there is no indication that IWRM implementation levels are greatest where they are most needed

3. The setup of the social aspect of this work needs to be clearer. After reading this revision of the manuscript, it seems that hotspots refer more to vulnerability to future problems, rather than impacts from past problems. Is this the intention? Is this analysis really showing hotspots of social and ecological “impacts from” freshwater stress (as in the title) or, rather, “vulnerability to” it? I’m confused because the social aspect in this analysis is measured specifically as the (“current” [i.e., 2015 in the Varis et al data set]) capacity of a society to adapt to future problems, which is then mapped onto a society’s vulnerability to the future problem of freshwater stress. Also, you repeatedly mention the potential for (future) impacts (Line 224 and 395), rather than actual (past) impacts. Thus, the title doesn’t seem to match the main body.

This is a very useful reflection, and we thank the reviewer for providing such a deeply engaged comment. It is correct that our study considers the potential for impacts (i.e. vulnerability) rather than summarizes already documented impacts. So, the reviewer is correct that our title (Hotspots of social and ecological impacts from freshwater stress and storage loss) may give the impression that our analysis focused on past impacts rather than potential for future ones. In response to this comment, we have made a number of minor (but, we believe, very meaningful) changes to the framing of the manuscript:

New title:

Hotspots for social and ecological impacts from freshwater stress and storage loss

- We think this new title better reflects our focus on the potential for future impacts (hotspots for impacts) while retaining the causal link that is core to our study (impacts from freshwater stress and storage loss)

Line 3: (in abstract)

Here, we focus on the potential for social and ecological impacts from freshwater stress and storage loss.

Line 28:

In this paper, we consider the potential for freshwater stress and storage loss to impact humans and ecosystems.

Line items

Line 1: Is this a run-on sentence? Consider separating “cycle however” with a period or semicolon.

Yes and we have revised. Updated text is already provided on this issue above.

Line 27: “global freshwater crisis” seems jarring because so far, you say only how people “influence”, “stress”, and even “dominate” the hydrological cycle, but that doesn’t necessarily lead to “crisis”. Below, on Line 161, you state the positive impacts of “stress”. Consider defining the crisis specifically (eg, lack of freshwater quantity, poor water quality, death?) or simply adding a reference.

Though we think there is sufficient evidence to frame it is a crisis, we are not particularly seeking to draw special attention to this claim and thus have reworded as:

Line 27:

... is crucial to confronting global freshwater challenges”, yet the latter has received considerably less attention.

Line 30: Consider rewording. Currently, it reads as “we consider impacts of xx and xx by synthesizing a subset of xx and xx datasets that exist with xx, xx, and xx datasets.”

We have reworded this sentence. See below.

Line 28--:

In this paper, we consider the potential for freshwater stress and storage loss to impact humans and ecosystems. We do this by synthesizing a subset of the few but critical global ecohydrological and sociohydrological datasets that exist with freshwater storage, freshwater withdrawal, and streamflow datasets (see Supplementary Table 1).

Line 54: Change “behvaiour” to “behaviour”.

Fixed

Line 55: Consider adding a comma between “research and.”

Done

Line 57: “these three fields” Which three? The sentence is long so I stopped counting the fields. Are they water scarcity, water security, and social-ecological systems research? It is unclear, given the way the previous sentence is written (the combination of social-ecological makes it seem like there can be four). Perhaps add a comma (see comment for Line 55) and/or add “additional” between “integrating concepts”.

We have restructured this sentence entirely so that each core objective now is allocated its own sentence.

Line 57--:

We combine concepts from these three fields to address the following core objectives of this study: (1) Assess the global co-occurrence of freshwater stress and freshwater storage trends at the basin scale. (2) Analyze the relationship between social adaptive capacity and ecological sensitivity indicators with freshwater stress and storage trends. (3) Derive the global gradient in social-ecological vulnerability to freshwater stress and storage trends by considering all indicators listed above, and identify hotspot basins as those with high vulnerability values with respect to the global distribution. (4) Evaluate current levels of integrated water resources management within hotspot basins.

Line 60: Change “tress” to “stress”.

Fixed – included in the updated text in the response immediately above.

Line 62: Consider that the total list (ie, global gradient) and the list of hotspots-only are separate

objectives. They are products that are useful in different ways. Otherwise, as it is, this sentence is too long and could be broken up with a comma before “and identify”.

We have added a comma before “and identify”.

Box 1 Freshwater stress: The second sentence (“The W/Q ratio...”) just restates the first sentence (“The ratio of...”). You can cut out the second sentence, keeping the first, and replace “evaluated” with “calculated”. Also, how about not using stress twice, so “We refer to basins with $W/Q \geq 10\%$ as stressed basins...”?

We agree that the previous definition was verbose and we have replaced it with:

Line 71 (inside Box 1):

The ratio of **annual** freshwater withdrawal (W) to annual streamflow (Q). We refer to basins with $W/Q \geq 10\%$ as stressed basins and those with $W/Q \geq 40\%$ as highly stressed basins.

Box 1 Freshwater storage trends: Please add the years of the trends here parenthetically and with reference to the Methods.

Done – see reply to comment on this issue earlier in this document.

Line 73: “We mapped freshwater stress and trends in freshwater storage” How about something like “We mapped current freshwater stress and past-to-current (c. 2002–2016) trends in freshwater storage” or something similar? It doesn’t really matter how you word it as long as you distinguish that the stress is a present state of things, whereas storage is a recently past trend over a period of multiple years. You can do this in Box 1 first, so you don’t have to change the subsequent text?

As we now specify the temporal range of the freshwater storage trends explicitly in Box 1, we believe the addition of additional terminology (e.g. ‘past-to-current’) may lead to additional confusion.

Line 85: Consider adding a comma between “stress as” and “stress while”.

Done

Line 94: The term “human activity” (Line 93) is sufficiently vague and opaque that it has no meaning. What specifically and literally does it mean? Typically, I would think of resource extraction, water re-routing, etc, and there are plenty of papers in the literature that you can reference. These specific things don’t need to be included in any analysis, but simply provided as background so the reader can understand why this analysis matters. Can you give one or two specific examples in parentheses, or at least put a reference (similar to how you put a reference for “natural variability” at the end of the sentence)?

We have modified our wording of this line to be more specific. As we use the freshwater storage trends derived by Rodell et al. (2018), we also used (in previous versions of this manuscript) their attribution terminology and classification system of freshwater storage trend drivers. However, we agree that the term ‘human activity’ is ambiguous, and so we clarify that the majority of the basins being discussed in this line coincide with heavily irrigated regions.

Line 94--:

Predominantly, these regions **are agriculturally significant and heavily irrigated⁹**, with the exception of a few basins in South America whose trends are likely the product of natural variability⁹.

Line 103: Great point.

Thank you

Line 115: This is an important point, but I think it gets lost because the sentence is so long. Consider breaking it into smaller sentences.

Done. See revised text below.

Line 116--:

We find less taxonomic biodiversity in the freshwater stressed and drying basins, and greater biodiversity in unstressed and wetting basins. Roughly the same number of wetlands of international importance are found in stressed and drying basins as in stressed and wetting basins.

Line 116: Consider adding a comma between “basins and”.

Done – see above.

Line 121: Consider adding a comma between “distributions as”. Again, this is an important point, but I think it gets lost because the sentence is so long. Consider breaking it into smaller sentences.

We have restructured this sentence accordingly. See below.

Line 119--

While these totals represent the magnitude of potentially affected biodiversity and wetlands, taxonomic biodiversity is only one of many critical facets of biodiversity²⁷, and freshwater stress and storage trends are but two of many variables impacting global biodiversity²⁸. Thus, we urge caution in interpreting the role of freshwater stress and storage in driving differences in these biodiversity distributions.

Line 133: Consider adding a comma between “coordinate and”, and adding a semi-colon or period between “coordinate with the” while cutting out “with”. The sentence is too long.

Done

Line 146: It would be appropriate to mention in a few words that you lumped social indicators by basin, then refer the reader to the Methods for further information. It is an abrupt transition from the hydrologic status of a basin, to the social (-adaptive) status of a (hydrologic) basin.

We agree. However, rather than add this clarification on line 146, we have added this in the preceding paragraph where we introduce the data and concept.

Line 128:

We mapped and analyzed the co-occurrence of freshwater stress and storage trends with an existing global dataset of social adaptive capacity²³ summarized at the basin scale (Fig. 2)

Line 158: Consider adding a comma between “development which”. It’s a very long sentence.

We have split this sentence in two (now line 148 with revisions and figures removed from the document).

Line 170: Consider breaking up this sentence (two “which” seems a bit much).

We have revised the structure of the sentence so that “which” only appears once. See below.

Line 159--:

One such example is Sustainable Development Goal (SDG) 6.4 (“reduce the number of people suffering from water scarcity”), which we argue should increasingly be linked to targets of SDG 10 (“reduce inequality within and among countries”).

Line 181: Capitalize “population”? Or change preceding period to comma?

We have removed the list of plotting dimensions from the legend to make consistent with the legend of Figure 1, which included the typo highlighted above.

Line 186: Perhaps, to drive the point home about what a hotspot is, consider changing this sentence to “We mapped the global gradient in social-ecological vulnerability to freshwater stress and storage loss at the basin scale and, from this, identified the hotspot basins of vulnerability”. “Basins of vulnerability” seems more in line with what you say (“vulnerability hotspots”) in Line 217. This list of hotspot basins is a top contribution of this paper but it gets a bit lost in the wordiness and repetition of the sentence, which starts with “gradient in social-ecological vulnerability” and ends with nearly the same thing (“global

vulnerability distribution”).

We think this is a very nice rewording. However, we also think that ‘hotspot basins of vulnerability’ could seem like yet another term for readers to reconcile (and deviates from our use of the term ‘hotspot basin’ in the key terminology box). We have revised this sentence, largely as suggested by the reviewer while avoiding the double use of ‘vulnerability’ in the sentence (as we highlight in yellow in the reviewer comment above).

Line 164--:

We mapped the global gradient in social-ecological vulnerability to freshwater stress and storage loss at the basin scale and, from this, identified those with the greatest vulnerability as hotspot basins.

Line 244: Change “identify the basins” to “are those”.

Done

Line 247: Start the sentence with “Identification of”?

Done, now starts with “The identification of ...” (now line 227).

Line 285: “levels across all levels...” Awkward repetitive phrasing. Consider rewording (also Line 273: “tracks IWRM implementation at the national scale to track...”; Line 170: “which aims to “reduce the number of people suffering from water scarcity”, which...”).

We have modified accordingly.

Line 242--:

For IWRM implementation data, we rely on the IWRM Data Portal³⁵ which tracks progress on SDG 6.5.1 (“IWRM implementation at the national scale”).

Line 297: So would you consider the hotspots (Afghanistan, Algeria, Argentina, Egypt, India, Iraq, Kazakhstan, Mexico, Somalia, Ukraine, Uzbekistan, and Yemen) that are vulnerable but are not receiving IWRM as being even hotter hotspots? Should this be mentioned in the abstract? Seems important.

We think that considering these basins as ‘even hotter hotspots’ could lead to possible confusion as IWRM implementation is not used in our derivation of social-ecological vulnerability or hotspot basin identification process.

However, we do believe the low implementation levels of IWRM in these basins should lead to greater prioritization amongst hotspot basins. We think mentioning this in the abstract would be a great addition but our abstract length is already at the maximum 150 word limit.

Line 317: Change “between” to “among”.

Done (now line 280)/

Line 323: Break up sentence or add commas. It’s too long.

We have added commas.

Line 348: What is “ref.50” referring to?

This reference call-out (Di Baldasserre et al. 2013) now is explicitly made in the text.

Line 363: Break up sentence. It’s too long.

We have split this sentence in two.

Supplementary Table 2: You switch between present and past tense multiple times (e.g., “we aggregated” and “we calculate” in the “Additional Preprocessing” of “Freshwater withdrawal...”; “we estimated” in the “Additional Preprocessing” of “Streamflow”). Pick one and be consistent.

We use only the past tense in the revised table.

General comments

-drying may be more commonly associated with drought than anthropogenic withdrawal. However, we also note that use of the term drying is also widely used in the context of human-driven storage loss

I thought, given that this journal is interdisciplinary and not limited to the context of human-driven storage loss (or even hydrology), using the word “drying” needed a clarification. Your use of it (and justification) is absolutely legitimate, but the reader can get confused. Including it in Box 1 is a great idea.

We are pleased that the addition of Box 1 provides useful clarification.

-we have removed ecological-only hotspot and social-only hotspot results but preserve the presentation of the underlying ecological and social vulnerability results in Figure 3e,f.

These updated figures are beautiful and informative.

Thank you!

-We have removed any references to water “have” and “have not” regions of the world and now simply refer to trends in freshwater storage in the context of existing freshwater stress explicitly.

-We have removed all use of the term “threat”

I think this makes your analysis clearer and more objective.

We agree - thank you for the suggestion.

-Thus, “hardships/limitations” (such as freshwater storage loss) can affect system resilience in future states through feedback mechanisms. If resilience were not a dynamic property, efforts to promote system resilience would not exist; however this is a major focus within social-ecological systems research.

I agree with this. My point was that a system can experience drying to a given tolerance without its resilience being affected. The problem is, as you say, feedback mechanisms can push a system past a tipping point into a “future state”, in which it loses/gains resilience. Is that what is happening here, though? Is the freshwater stress within the tolerance of these environmental systems or not? Is it pushing these systems past their functional boundaries into new states in which their resilience/function changes? I didn't see any language about future states, tipping points, non-stationarity, altered resilience, or the like.

As I understand it, you are showing areas of freshwater stress and then making further implications about resilience in these areas. So, the use of “resilience” was unclear in the first draft. The language (e.g., Lines 315 and 388 in the first draft) in the first draft sounded like it linked the freshwater stress from this study to resilience generally. Here in the second draft, I think you clarify in Lines 363 and 394 that evaluating the resilience of your study basins is outside the scope of this study.

We are glad that lines ~363 and ~394 in the revised manuscript provide the necessary clarification.

Supplementary Line 126:

It seems the reviewer accidentally deleted this comment. Perhaps this comment was referring to our use of feed back as a verb, which is clarified in the comment below.

-Line 145: “feedback” is two words

It can also be written as one word:

<https://dictionary.cambridge.org/dictionary/english/feedback>

You were using it as a verb, though (“Ecological impacts can also feedback to affect freshwater stress...”).

<https://dictionary.cambridge.org/dictionary/english/feed-back>

I see you changed it in the revision (“Ecological impacts can also feed back...”; Line 126, Supplementary Information). Also, you have it written correctly in Line 131, Supplementary Information.

We have ensured that we use “feed back” wherever it is used as a verb.